# Logical rotation of non-separable states via uniformly self-assembled chiral superstructures

Yi-Heng Zhang[1], Si-Jia Liu[1], Peng Chen ●[1] ✉, Dong Zhu[1], Wen Chen[1], Shi-Jun Ge[1], Yu Wang ●[1], Zhi-Feng Zhang[1] & Yan-Qing Lu ●[1] ✉

The next generation of high-capacity, multi-task optical informatics requires sophisticated manipulation of multiple degrees of freedom (DoFs) of light, especially when they are coupled in a non-separable way. Vector beam, as a typical non-separable state between the spin and orbital angular momentum DoFs, mathematically akin to entangled qubits, has inspired multifarious theories and applications in both quantum and classical regimes. Although qubit rotation is a vital and ubiquitous operation in quantum informatics, its classical analogue is rarely studied. Here, we demonstrate the logical rotation of vectorial non-separable states via the uniform self-assembled chiral super-structures, with favorable controllability, high compactness and exemption from formidable alignment. Photonic band engineering of such 1D chiral photonic crystal renders the incident-angle-dependent evolution of the spatially-variant polarizations. The logical rotation angle of a non-separable state can be tuned in a wide range over $4\pi$ by this single homogeneous device, flexibly providing a set of distinguished logic gates. Potential applications, including angular motion tracking and proof-of-principle logic network, are demonstrated by specific configuration. This work brings important insight into soft matter photonics and present an elegant strategy to harness high-dimensional photonic states.

The vectorial nature of light leads to optical fields with complex polarization structures[1,2]. Vector beam (VB), a representative example of which is characterized by spatially-variant linear polarizations in its transverse plane[3,4], has found its applications in diverse areas of optics and photonics, including communication[5], super-resolution imaging[6], and laser processing[7]. VB is the coherent superposition of spin and orbital angular momentum (SAM and OAM) eigenstates[8–10], and can be described by the same SU(2) algebra of an entangled system[11]. Featuring the non-separability between the spin and orbital degrees of freedom (DoFs)[12,13], such vectorial state is conceptualized as a pair of non-separable cebits[14,15], as a classical analogue of two entangled

qubits. On one hand, this similarity enables the applicability of many quantum methodologies to exploit compelling properties of classical VBs, contributing to noise-tolerant communication[4] and kinematic sensing[16]. On the other hand, vectorial non-separable states pave the road to the easier implementation of some quantum protocols[13], which are challenging at the single photon level, including teleportation[17] and entanglement beating[18]. The above systems are generally built on logical operation modules, such as CNOT gate and projective measurement, which have been physically realized by wave plates, prisms, and spatial light modulators[15–17]. However, the logical rotation of vectorial non-separable states has been seldom studied, which is the

[1]National Laboratory of Solid State Microstructures, Key Laboratory of Intelligent Optical Sensing and Manipulation, College of Engineering and Applied Sciences, and Collaborative Innovation Center of Advanced Microstructures, Nanjing University, 210093 Nanjing, China. ✉e-mail: chenpeng@nju.edu.cn; yqlu@nju.edu.cn

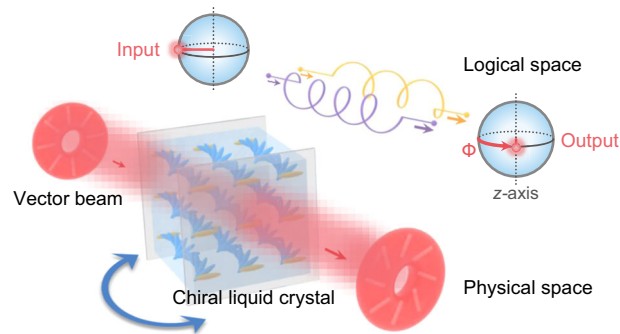

**Fig. 1 | Chiral superstructures mediated logical rotation gate for vectorial non-separable states.** The logic gate enabled by uniform left-handed CLC superstructures is schematically illustrated, which is mechanically rotated to form various incident angles, causing simultaneous rotation of local linear polarizations of the incident VB. In logical space (i.e., the higher-order Poincaré sphere), this physical process corresponds to a logical rotation $\Phi$ of the vectorial non-separable state around the $z$-axis. The yellow/purple spiral lines symbolize the propagating eigenmodes with different phase retardations in CLC.

classical analogue of qubit rotation, an essential and universal operation in quantum informatics[19,20].

Self-assembled chiral superstructures are wonders in nature. Those delicate chiral structures can be employed to induce fancy light-matter interactions beyond traditional optical devices[21–23]. Among all, chiral liquid crystal (CLC) is well known as a high-efficient and easy-manufactured building block. Anisotropic liquid crystal molecules spontaneously organize into helices, forming the 1D chiral photonic crystal with the spin-selective photonic band gap (PBG)[24–26]. The spin state with the same chirality as that of the helical structure is reflected within this PBG[27], while the opposite one is transmitted. Collaborated with multiple stimuli responsiveness, CLCs have fueled remarkable advances in the fields of programmable structural colors[28,29], tunable lasers[30], augmented/virtual reality displays[31], and dynamic beam shaping[32,33]. Unfortunately, when the VB meets the PBG, the spatial separation of two orthogonal spin states will destruct the pivotal non-separability of the non-separable state, preventing its further applications in optical informatics.

In this work, we turn to the vicinity of PBG (i.e., the region outside and near the PBG) where both spin states can propagate forward, and theoretically reveal that the non-separability of the input vectorial cebits is well protected. Moreover, the special photonic band of CLC renders the incident-angle-dependent evolution of non-separable states. By using the uniformly self-assembled chiral superstructures, we further experimentally demonstrate the logical rotation of VBs around the $z$-axis in the logical space (Fig. 1). The optical rotation induced by CLCs is nearly independent of the polarization structures of the input state, and this manipulation on the spin DoF will be coupled into the orbital DoF due to their intrinsic non-separability. As a result, our uniform chiral structures are sufficient here, exempted from the stringent requirement of alignment in common structured light control. By mechanically rotating the CLC device to specific angles, it can be transformed into different logic gates, including S gate, T gate, Pauli-Z gate, and projection operator. Moreover, our proposed device enables angular motion tracking, and serves as a potential building block for the logic network of non-separable states. This study demonstrates a practical scheme to tailor non-separable states in a dynamic and multi-functional way.

## Results
### Principles
The VB is a non-separable superposition of tensor products of SAM and OAM states. As an example without any loss of generality, the radially polarized VB can be written as the following form (normalization constants are omitted)[4]:

$$|\psi\rangle = |-\sigma\rangle|+l\rangle + |+\sigma\rangle|-l\rangle. \qquad (1)$$

$\sigma = 1$ and $|\pm\sigma\rangle$ denotes the left- and right-handed circular polarization (LCP and RCP), while $l = 1$ and $\pm l$ denotes the OAM topological charge[8,9]. This pair of basis vectors span a two-dimensional Hilbert space, the so-called higher-order Poincaré sphere (HOPS)[34] (i.e., corresponding logical space), and $|\psi\rangle$ is a maximally non-separable state located at the equator (Fig. S1 and Supplementary Note 1)[12,13]. By virtue of non-separable spin and orbital DoFs, the VB can usually be treated as a quantum-like state, akin to Bell states in two qubit system. It is worth noting that classical non-separability is a local correlation between different DoFs, distinguished from the non-local quantum entanglement.

Inspired by the qubit rotation, we focus on the logical rotation of such classical non-separable state around the $z$-axis of its logical space, namely the axis perpendicular to the equatorial plane of the HOPS. Such logical rotation operator can be expressed as the tensor product form:

$$\mathbf{O}_{\text{spin}} \otimes \mathbf{I}_{\text{orb}} = (|+\sigma\rangle\langle+\sigma|+e^{i\Phi}|-\sigma\rangle\langle-\sigma|) \otimes (|+l\rangle\langle+l|+|-l\rangle\langle-l|). \qquad (2)$$

Operator $\mathbf{O}_{\text{spin}}$ acts on the spin DoF, resulting in a phase difference $\Phi$ between the two orthogonal SAM eigenstates, while $\mathbf{I}_{\text{orb}}$ is an identity operator on the orbital DoF. If this unitary and one-sided operator acts on the state $|\psi\rangle$ in Eq. (1), the output state is

$$|\psi_{\text{out}}\rangle = \mathbf{O}_{\text{spin}} \otimes \mathbf{I}_{\text{orb}}|\psi\rangle = e^{i\Phi}|-\sigma\rangle|+l\rangle + |+\sigma\rangle|-l\rangle. \qquad (3)$$

The output state is rotated by an angle $\Phi$ around the $z$-axis of the same HOPS (namely its logical space), which corresponds to the synchronous rotation of an angle $\Phi/2$ for local linear polarization of the input VB. If $\Phi$ can be regulated by certain external factor, a variety of logic gates will be realized with dynamic controllability. For instance, when $\Phi = \pi$, the initial radially polarized beam will be transformed into an azimuthally polarized beam, behaving as the classical counterpart of Pauli-Z gate[20]. Meanwhile, if $\Phi$ can be controlled by some intrinsic dimensions of the photonic state, the logical rotation gate will be analogous to the CNOT gate, one fundamental component of quantum logic circuits[19]. Such control mode would pave the way for a complete logic network of non-separable states.

For normal incidence, the CLC helices of pitch $p$ (Fig. 2a) give rise to a PBG with a wavelength range of $n_o p - n_e p$, where $n_o/n_e$ is the ordinary/extraordinary refractive index, respectively[27]. The incident light wave stimulates certain composition of eigenmodes in CLC based on mode matching conditions at the boundary (Fig. 2a)[24]. Generally, a pair of forward-propagating eigenmodes with nearly-orthogonal polarizations will be excited, except that within the PBG, the one with the same chirality as the CLC will be reflected. While for the oblique incidence at an angle $\theta$, the propagation dynamic of light resembles the normal incidence with a longer 'effective wavelength' $\lambda_{\text{eff}} = \lambda / \cos\theta$. The phase retardation between these two eigenmodes leads to the rotation of the polarization direction, quantified by the optical rotatory power[25,35,36]. Eigenmodes also determine the polarization of the output superimposed state, evaluated by the polarization figure of merit (PFM) (see details in Methods)[25].

The photonic band structure of a left-handed CLC is systematically investigated by analytical calculation (Supplementary Note 2). For linearly polarized normal incidence, the optical rotatory power and the PFM are plotted against wavelength (Fig. 2b). Notably, within the vicinity of the PBG, the PFM is relatively high, indicating the superimposed linearly polarized states. Moreover, the corresponding optical rotatory power is strongly dispersive. Since the oblique incidence

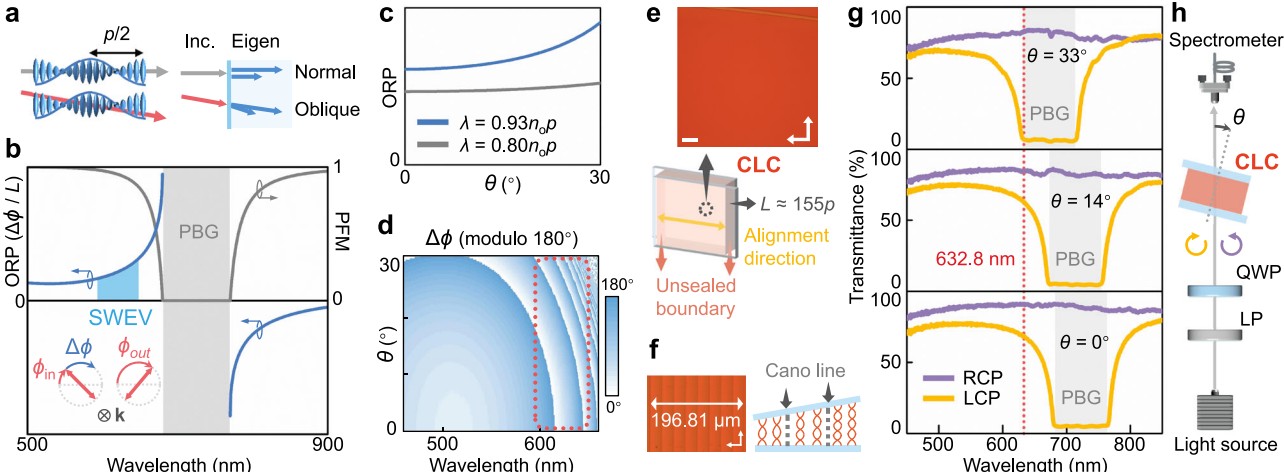

**Fig. 2 | Design, configuration and spin-selective PBG of the uniform chiral superstructures. a** Schematic of normal (gray arrow) and oblique (red arrow) light propagating in the chiral superstructures of pitch $p$. Blue rods denote LC molecules. Certain composition of optical eigenmodes (blue arrows) in CLC is stimulated according to the boundary conditions. **b** Analytical optical rotatory power (ORP, blue curve) and polarization figure of merit (PFM, gray curve) as functions of the wavelength for linearly polarized normal incidence at left-handed CLC. PBG, photonic band gap; SWEV, short-wavelength edge vicinity; $L$, interaction length. Inset: the input ($\phi_{in}$) and output ($\phi_{out}$) linear polarization direction, and the polarization rotation ($\Delta\phi$). k denotes the wave vector. **c** Theoretical dependence of ORP on the incident angle $\theta$, for the incident wavelength within the short-wavelength edge vicinity of the PBG ($\lambda = 0.93n_op = 633$ nm, blue curve) and away from the PBG ($\lambda = 0.80n_op = 544$ nm, gray curve), respectively. **d** Simulated

dependences of $\Delta\phi$ on the wavelength and $\theta$ for $\phi_{in} = 0°$, with the SWEV labeled by red dotted circle. **e** Schematic of the CLC device, and the cross-polarized reflective optical micrograph of the region indicated by a black dotted circle. The alignment direction (yellow arrow) and the unsealed boundaries (red arrows) of the LC cell are labeled, respectively. $L$ is the cell gap. The scale bar is 100 μm. **f** Micrograph and schematic of the CLC helices in a wedge cell. The total height of local helix alters by $p/2$ across each two adjacent Cano lines. **g** Measured transmittance spectra of the CLC device under RCP (purple) and LCP (yellow) incidence at various incident angles $\theta$. The PBG and the operating wavelength 632.8 nm are labeled. **h** Schematic of the optical setup for measuring spectra (**g**). LP, wire-grid linear polarizer; QWP, achromatic quarter-wave plate. The purple anticlockwise and yellow clockwise arrow denote RCP and LCP, respectively.

resembles the normal incidence with a longer effective wavelength, the polarization rotation angle $\Delta\phi$ (i.e., $\Phi/2$) would be tuned by altering the incident angle $\theta$. To verify this supposition, both the perturbation method (Supplementary Note 2 and Fig. 2c) and numerical simulation[37] (Supplementary Note 3, Fig. 2d and Figs S2, S3) are performed. As expected, in the short-wavelength edge vicinity of the PBG, $\Delta\phi$ can be tuned in a range over π (i.e., at least one cycle) by varying $\theta$ (Fig. 2d). Furthermore, PFM ≥ 0.85 indicates acceptable linearly polarized output state (Fig. 2b and Fig. S2a). This is because the polarization states of the propagating eigenmodes in CLC mostly accord with the SAM eigenstates (Fig. S3). These behaviors satisfy the requirement of $O_{spin}$ by introducing certain relative phase between $|-\sigma\rangle$ and $|+\sigma\rangle$. But for the long-wavelength edge vicinity of the PBG, the output states rapidly deviate from linear polarization when $\theta$ increases (Fig. S2b), with PFM < 0.85 for $\theta > 15°$, making it incompetent to logically rotate vectorial non-separable states.

In addition to the operation on spin DoF, the CLC superstructures with uniform standing helices would not affect the orbital DoF, satisfying the desired behavior of $I_{orb}$. To sum up, for incident light within the short-wavelength edge vicinity of PBG, the uniform CLC can act well as the proposed logical rotation gate (see detailed analysis in Supplementary Note 4), whose functionality can be flexibly controlled by the relative angle between the CLC device and the incident light. One straightforward control mode is through the mechanical rotation of the CLC device. As illustrated in Fig. 1, the local polarizations of the VB synchronously rotate during the propagation through the inclined CLC. In corresponding logical space, the CLC logic gate shifts the non-separable state around the $z$-axis, whose logical rotation angle $\Phi$ can be tuned in a wide range, potentially acting as various logic gates.

## Uniformly self-assembled chiral superstructures
To experimentally verify the theoretical prediction above, nematic LC host E7 was doped with left-handed chiral dopant S5011 (see respective molecule structures in Fig. S4) to get a stable, low-cost and easily

fabricated CLC material. With a carefully designed concentration of S5011, the short-wavelength edge vicinity is adjusted to include the intended wavelength 632.8 nm. The cell gap $L$ (i.e., the interaction length) is chosen to be as thick as 70 μm to produce sufficient dynamic phase retardation. Based on the photosensitive azo-dye SD1[38], the surface alignment is set perpendicular to the unsealed boundaries of the LC cell. After filled with the prepared CLC mixture at 80 °C, the cell is procedurally cooled to room temperature and kept in the dark for 3 days. In this process, owing to the system's tendency to minimize the free energy, the oily streak defects gradually merge and sufficiently self-assemble into uniform standing helices (see details in Methods and Fig. S5). Afterwards, the designed uniform chiral superstructures are achieved, and their homogeneous planar textures are shown in Fig. 2e and Fig. S5, whose functional region can be scaled up to about 5.8 cm² (Fig. S5). Grandjean-Cano wedge measurement indicates that $p = 450$ nm and about 155 pitches are involved along the helix axis across the whole cell (see Fig. 2f and Methods).

To reveal the spin-selective PBG, the transmittance spectra for different circular polarization inputs (Fig. 2g) are measured by the setup in Fig. 2h. The incident angle $\theta$ is defined against the central normal axis of the CLC device. For the left-handed chiral superstructures, the SAM eigenstate $|+\sigma\rangle$ (LCP) is stopped/reflected within the PBG, while $|-\sigma\rangle$ (RCP) can propagate and transmit. When the CLC device tilts from the normal, the short-wavelength edge of PBG blue shifts from 680 nm ($\theta = 0°$) to 670 nm ($\theta = 14°$). Finally, when $\theta$ reaches 33°, the PBG covers the operating wavelength of 632.8 nm. In this case, two SAM eigenstates will be separated, indicating that CLC device no longer behaves as $O_{spin}$. In addition, we further study the dependence of the polarization rotation on $\theta$ using the setup in Fig. 3a. When the incident polarization direction is fixed at $\phi_{in} = 90°$, the co-/cross-polarized transmittance spectra (Fig. 3b) exhibit substantial oscillations and numerous transmittance extremes, clearly showing the wavelength-dependent variation of the output polarization direction $\phi_{out}$. Furthermore, by rotating the analyzer sequentially from 0° to

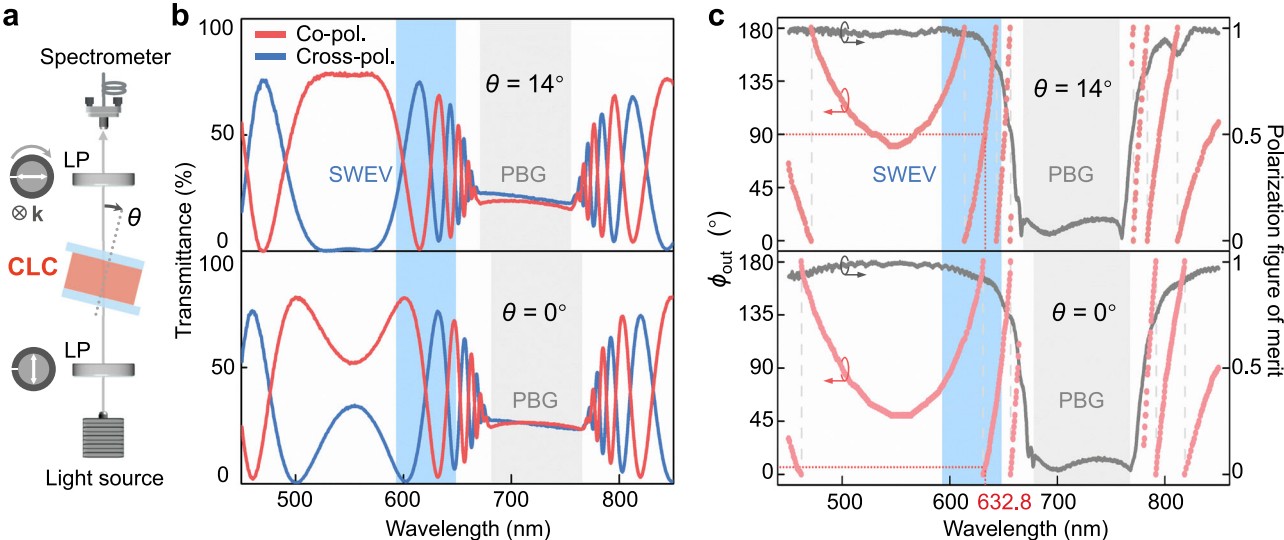

**Fig. 3 | Polarized transmittance spectra of the chiral superstructures.**
**a** Schematic of the optical setup for measuring polarized transmittance spectra. LP, wire-grid linear polarizer. The fixed polarizer ($\phi_{in} = 90°$) and the clockwise rotating analyzer are illustrated. k denotes the wave vector. **b** Measured co-polarized (red) and cross-polarized (blue) transmittance spectra of the CLC device at the incident angle $\theta = 0°$ and 14°, respectively. **c** Spectrally resolved output polarization direction $\phi_{out}$ (red scatters) and PFM (gray curves) at $\theta = 0°$ and 14°, respectively. As marked by horizontal and vertical red drop lines, the increment of $\phi_{out}$ approaches 90° at 632.8 nm. The PBG and the short-wavelength edge vicinity (SWEV) are labeled by gray blocks and blue blocks, respectively.

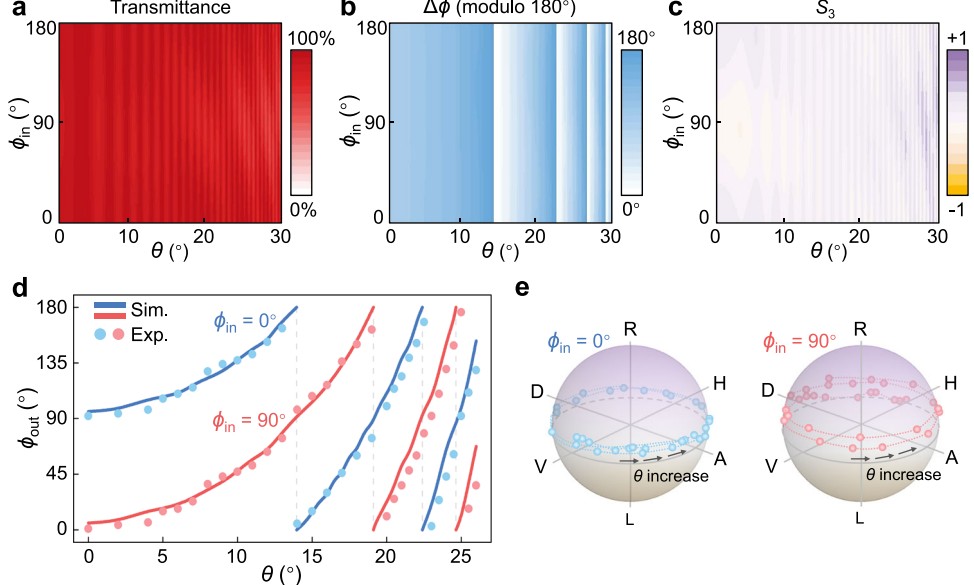

**Fig. 4 | Logical rotation of spin cebits at 632.8 nm. a–c** Simulated dependences of (**a**) the transmittance without analyzer, (**b**) the polarization rotation $\Delta\phi$, and the Stokes parameter $S_3$ on the incident angle $\theta$ ranging from 0° to 30° and the input polarization direction $\phi_{in}$ ranging from 0° to 180°. **d** Dependence of the output polarization direction $\phi_{out}$ on $\theta$, when the input is horizontal ($\phi_{in} = 0°$, blue) and vertical ($\phi_{in} = 90°$, red) linear polarization, respectively. Scatters and curves denote the experimental and simulated results, respectively. **e** Measured Stokes trajectory of the output state on the Poincaré sphere with increasing $\theta$ for horizontal (blue) and vertical (red) linear polarization incidence, respectively. H, D, V, and A correspond to $\phi_{out} = 0°$, 45°, 90°, and 135°, respectively.

180° and combining all transmittance spectra, we extract the spectrally resolved $\phi_{out}$ and PFM (see Methods). Comparing the two points labeled by horizontal and vertical drop lines in Fig. 3c, when $\theta$ grows from 0° to 14°, the increment of $\phi_{out}$ at 632.8 nm approaches 90° with the PFM > 0.90, indicating that the polarization rotation can be flexibly tuned by altering the incident angle.

## Logical rotation of spin cebits

A linearly polarized input state at 632.8 nm is applied to verify the operation of this CLC logic gate on the pure spin DoF (i.e., the spin cebits). The output state is theoretically simulated with $\phi_{in}$ varying

from 0° to 180° and $\theta$ from 0° to 30°. The transmittance without analyzer, the polarization rotation $\Delta\phi$, and the Stokes parameter $S_3$ are calculated and displayed in Fig. 4a–c, respectively (see $\phi_{out}$ in Fig. S6). The transmittance is relatively high for a wide range of $\phi_{in}$ and $\theta$. Particularly, $\Delta\phi$ is almost invariant for different $\phi_{in}$, and evidently depends on $\theta$. The $S_3$ graph shows PFM > 0.90 for $0° \leq \theta \leq 26°$ (Methods), implying the favorable linear polarization of the output state. All these properties again prove that the requirements of $\mathbf{O}_{spin}$ are well fulfilled. In experimental verification, two typical cases of $\phi_{in} = 0°$ and 90° are exhibited in Fig. 4d. With $\theta$ rising, $\phi_{out}$ increases correspondingly and covers a wide range over 360°. Accordingly, the measured

output states are positioned on the standard Poincaré sphere (Fig. 4e), namely the logical space of spin cebits. The input states are at the equator and their evolution trajectories depict the incident-angle-controlled logical rotation of spin cebits with $\Phi$ over $4\pi$. These experimental results match well with numerical simulations.

## Logical rotation of vectorial non-separable states

Figure 5a illustrates the setup to perform the logical rotation of vectorial non-separable states by rotating the CLC device around the vertical axis (Fig. 5a right-inset). The initial non-separable state $|\psi\rangle$ in Eq. (1) with radial polarization (Fig. 5a left-inset) is generated by a nematic LC $q$-plate[39] at 632.8 nm (Methods). The output transformed states are analyzed by the intensity profiles transmitted through a horizontal analyzer, and verified by the polarization map from the Stokes measurements (Methods). When the CLC device is tilted at $\theta = 14.0°$, the analyzed intensity and polarization distributions (Fig. 5b) both coincide with those of the input state, implying that an identity operator is achieved at this incident angle. By setting $\theta = 15.5°$, $17.0°$, and $19.1°$, the analyzed intensity profile with two separated lobes rotates $22.5°$, $45°$, and $90°$, respectively (Fig. 5c–e). Corresponding polarization maps indicate that local polarizations rotate simultaneously and remain linear in general, matching well with the results of pure spin cebits in Fig. 4. Fulfilling the rotation operator proposed in the Principles section, the logical rotation angle turns to $\Phi = \pi/4$, $\pi/2$, and $\pi$. These implement the classical counterparts of three vital quantum gates, T gate, S gate, and Pauli-Z gate[20], respectively. The evolutions of these vectorial non-separable cebits around the $z$-axis of the HOPS are also illustrated in Fig. 5b–e. The transmission efficiency is about 80% and can be further improved to over 90% using anti-reflection substrates (Fig. S7). Not limited to $l = +1$, we also execute the logical rotation of the non-separable states with $l = -1$ and $l = +4$ (Fig. S1b, c), and obtain a similar evolution trajectory on corresponding HOPSs (Fig. S8), which indicates the universal applicability of the proposed CLC device.

Meanwhile, when $\theta$ reaches $33.0°$, the VB with $l = +1$ is spin-selectively separated, because 632.8 nm is now included by the chiral PBG (Fig. 2g). The spin and the orbital DoFs are respectively examined by a circular polarization analyzer and a cylindrical lens (Methods). As shown in Fig. 5f, the original non-separable state is wholly decomposed into two spatially-separated tensor product states, namely $|-\sigma\rangle|+l\rangle$ in the transmission channel, and $|+\sigma\rangle|-l\rangle$ in the reflection channel, which becomes $|+\sigma\rangle|+l\rangle$ because of the reversed propagation direction and the conserved spin chirality[40,41]. Evidently, in this condition, the chiral superstructures act as a projection operator, instead of the logical rotation gate (Supplementary Note 4). All these experimental results conform with theoretical predictions and agree well with numerical simulations. In a word, we demonstrate the logical rotation for vectorial non-separable states merely using uniform chiral superstructures, and diverse functions can be flexibly altered by tilting this single device.

## Potential applications for angular motion tracking and logic network

In the above implementation, the logical rotation is controlled by the tilt angle $\theta_{mech}$ of the device (an external mechanical quantity), and such control mode is referred to as *mechanical mode* (Fig. 6a). $\theta_{mech}$ determines the output non-separable state as shown in Fig. 5b–e, which is characterized by the orientation angle $\beta$ of the dark fringe segmenting the analyzed intensity profile (i.e., the intensity minima). $\beta$ can be extracted via the digital image processing (Fig. S9). Notably, this quantitative $\beta$-$\theta_{mech}$ relation (Fig. 6b) can be applied for angular motion tracking. Mimicking the concept of sinusoidal vibration in mechanical engineering, an angular motion formulated as $\theta_{mech} = 21 + \cos(\Omega t) + \cos(2\Omega t)$ (unit: degree) is performed, possessing a 'fundamental' and a 'second harmonic' component. Based on the collected

intensity profiles and the working curve (Fig. 6b), the rotation trajectory of the device is reconstructed in Fig. 6c, which matches well with the preset vibration. Compared to traditional interferometric instruments, this scheme is based on a single optical path with high compactness and in situ manner, potentially executed with integrated laser source and image sensor.

A step from the single CLC logical rotation gate towards the complete logic network is vital for developing algorithms or protocols based on non-separable states. Accordingly, the logical rotation needs to be controlled by a photonic state, and the output state of the former stage needs to be inserted into the control and the input of the successive logical rotation gate[19,42] (see detailed disscussions in Supplementary Note 5 and Note 6). To satisfy these two requirements, on one hand, we introduce the photonic wave vector direction $\theta_{phot}$, an intrinsic DoF of light, as another control method of the logical rotation (Fig. 6d), and name it *wave vector mode* distinguished from the previous *mechanical mode* (see detailed analysis in Supplementary Note 5 and Fig. S10). Considering the relation $\theta = \theta_{mech} \cdot \theta_{phot}$, the altering $\theta_{phot}$ acts as the control, while the preset constant $\theta_{mech}$ plays the role of 'bias'[42] to offer the desired operating region of the logical rotation. A circuit (Fig. S11) to validate the *wave vector mode* is executed by the setup in Fig. 6e. Through the equal-energy distributed diffraction from a Dammann grating[43], the initial non-separable state with radial polarization is simultaneously encoded with $\theta_{phot} = -2.34°$, $-0.78°$, $+0.78°$, and $+2.34°$ in four diffracted channels (Methods). For the CLC logic gate tilted to a fixed $\theta_{mech} = 17.84°$, different intensity profiles of the analyzed outputs (Fig. 6f and Fig. S11) indicate that the logical rotation is efficiently controlled by the photonic wave vector. On the other hand, a reverse-controlling unit is demanded to reversely control $\theta_{phot}$ based on the non-separable state. CLC logical rotation gates and reverse-controlling units may assemble sequentially into a complete logic network (see configuration in Supplementary Note 6 and Fig. S12). In the proof-of-principle experiment, we consider the reduced non-separable states with $l = 0$ (i.e., the spin cebits), and a simplified logic network has been demonstrated with additional easily-accessible optical elements (Fig. S12). For more general non-separable states, versatile metasurfaces capable of manipulating multiple VBs[44,45] are promising candidates for the reverse-controlling unit. The composed functional logic network would have great potential in high-dimensional photonics, optical computing and high-capacity optical informatics. In particular, it might benefit some computation concepts in the analogous version of quantum walks[46,47], since the non-separable-states-mediated coins could allow more complex form and more flexible coin rotation rules in this logic network (see more discussions in Supplementary Note 6).

## Discussion

Main structural factors affecting the performance of the logical rotation have been deeply investigated, including the interaction length $L$, the ratio of operating wavelength $\lambda$ to the short-wavelength edge of PBG $n_o p$, and the chirality of CLCs. Firstly, sufficient $L$ is necessary for the functionality of the logical rotation gate, as proved by the highly restricted performance of a much thinner device in Fig. S13. Secondly, when $\lambda/(n_o p)$ increases, $\lambda$ gets closer to the PBG and the dispersion of optical rotatory power becomes stronger. Therefore, the logical rotation angle $\Phi$ depends more sensitively on $\theta$, while their dynamic ranges are also affected (Fig. S14). Thirdly, a CLC device with the opposite chirality leads to an opposite direction of the logical rotation (Fig. S15). Additionally, opposite incident angle $-\theta$ is equivalent to $+\theta$ because of the system symmetry[37]. In a word, for a specific application, a competent CLC logic gate can be carefully optimized in the light of the above principles, and the intended chiral superstructures can be fabricated readily with the proper type and concentration of chiral dopants, thanks to their intriguing self-assembly. Moreover, if the fantastic stimuli-responsive CLC materials are adopted, the dynamic

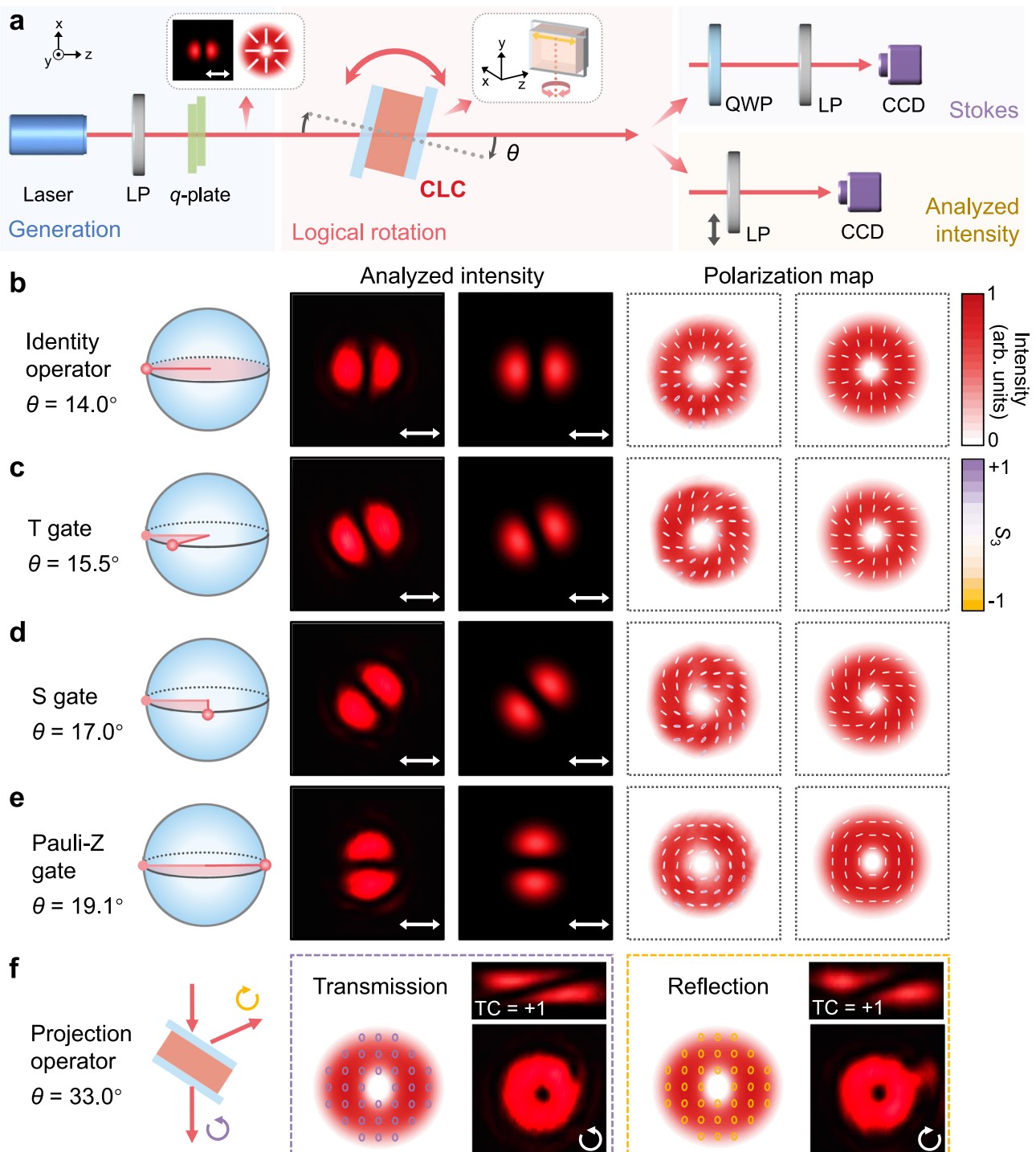

**Fig. 5 | Logical rotation of vectorial non-separable states at 632.8 nm. a** The optical setup includes four stages: generation of the initial non-separable state, its logical rotation via the proposed CLC logic gate, Stokes measurement, and analyzed intensity of the output state. LP linear polarizer, QWP quarter-wave plate, CCD charge-coupled device, Left-inset: simulated analyzed intensity profile and polarization map of the initial state; Right-inset: schematic illustration of the mechanical rotation of the CLC device. The x-axis is horizontal and the y-axis is vertical. **b−e** Schematic illustrations of the state evolution on HOPS, measured and simulated results of analyzed intensity profiles and polarization maps, when the CLC device performs (**b**) an identity operator, (**c**) a T gate, (**d**) an S gate, and (**e**) a Pauli-Z gate, at $\theta = 14.0°$, $15.5°$, $17.0°$, and $19.1°$, respectively. **f** Simulated polarization maps, measured intensity profiles through circular polarizer and OAM measurement by a cylindrical lens for the transmission and the reflection channels, respectively, when the CLC device performs the spin-selective projection at $\theta = 33.0°$. TC, topological charge. Polarization maps are shown as false-color intensity, and polarization ellipses are colored according to Stokes parameter $S_3$. The incident, reflected and transmitted light are illustrated on the left. The double-ended/clockwise/anticlockwise arrow denotes linear polarization/LCP/RCP, respectively.

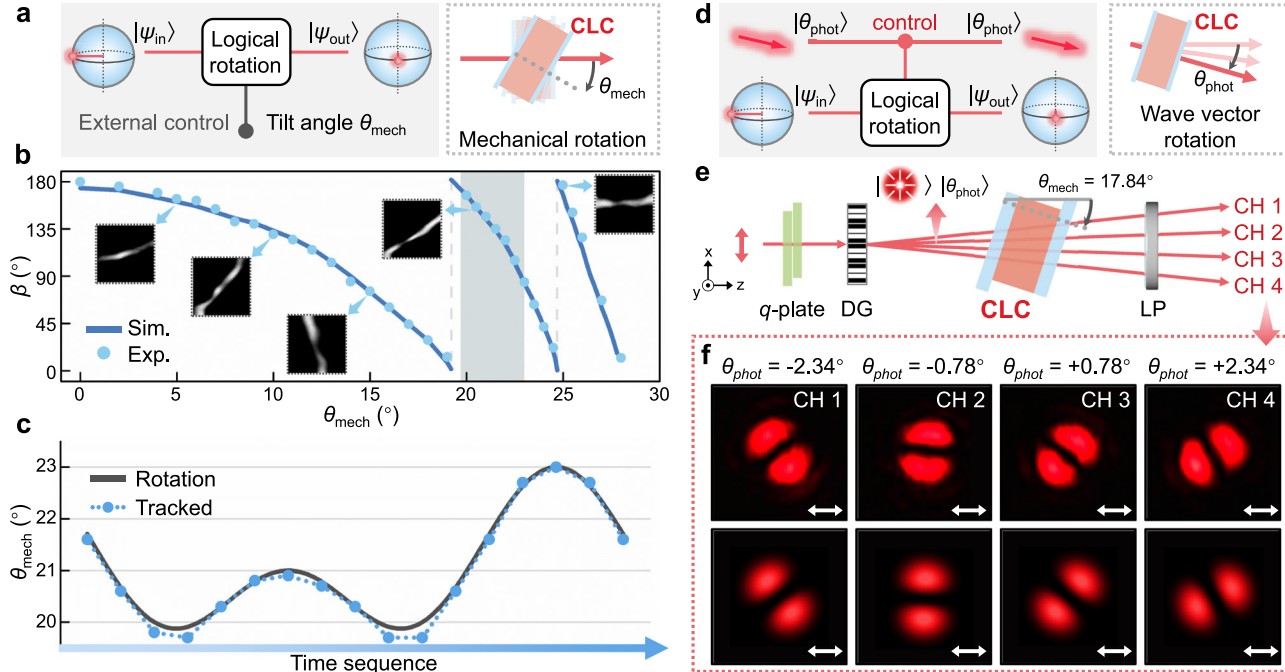

**Fig. 6 | Two control modes of the logical rotation of non-separable states and their potential applications. a** Circuit and physical implementation of the *mechanical mode*, where one non-separable cebit is accepted as an input, and an output cebit is determined by the external physical quantity $\theta_{mech}$. **b** Relation between $\theta_{mech}$ and $\beta$ based on experiment (scatters) and simulation (curves), with some grayscale-inversed stripes labeled correspondingly. Gray block denotes the operating region in the angular motion tracking. **c** Preset angular rotation (gray solid curve) and measured tracking trajectory (blue dotted curve) of the CLC

device. **d** Circuit and physical implementation of the *wave vector mode*, where an input non-separable cebit evolves to an output under the control of the wave vector (i.e., $\theta_{phot}$), an intrinsic DoF of light. **e** Optical setup for controlling logical rotation by wave vector direction. DG, Dammann grating; LP, linear polarizer; CH, channel. The initial radially polarized VB is encoded with $\theta_{phot} = -2.34°$, $-0.78°$, $+0.78°$, and $+2.34°$ in four channels, respectively. $\theta_{mech} = 17.84°$. **f** Measured and simulated results of analyzed intensity profiles of four channels at different $\theta_{phot}$. White arrows denote horizontal analyzers.

logical rotation controlled by light[29] or heat[10] could be rationally anticipated (Fig. S16).

In summary, we demonstrate the unprecedented logical rotation for vectorial non-separable states via uniform chiral superstructures. A variety of distinguished logic gates are well performed with favorable incident-angle controllability, which contributes to angular motion tracking in the field of metrology and potential logic network of non-separable states. Compared to rotators based on magneto-optic effect[48], our scheme enables a much wider tuning range and simpler configuration. Although angle dependence usually manifests in the undesirable form of energy loss or aberration[49] for traditional devices, here we unlock the incident angle as an extra DoF for dynamic controllability. Such control principle averts repetitive re-organization of the internal nanostructures and exhibits merits of synchronous response and high reliability. In conventional view, inhomogeneous optical elements are indispensable for tailoring structured light, such as spatially-variant waveplates or artificially-arranged sub-wavelength scatterers. Here, by fully exploiting the unique photonic band of chiral superstructures and the non-separability between the spin and orbital DoFs, we present a simple strategy to manipulate complex vectorial fields, which offers a glimpse into structured photonics free from formidable alignment in the real space. Though being classical, such logic gate has great potential to operate on single-photon or entangled states in quantum informatics as well. This work facilitates vigorous advances in soft matter photonics, and supplies an elegant solution for the exploitation and application of high-dimensional photonic states.

## Methods
### Materials
Photoalignment agent SD1 (Dai-Nippon Ink and Chemicals, Japan) was dissolved in dimethylformamide at a concentration of 0.35 wt %. The

left-handed CLC was prepared by mixing the nematic LC E7 (HCCH, China) and 2.25 wt % left-handed chiral dopant S5011 (HCCH, China). While for the comparative experiment of larger $\lambda/(n_o p)$ in Fig. S14, the dopant ratio was 2.28 wt%. The right-handed CLC in Fig. S15 was fabricated by mixing E7 and right-handed chiral dopant R5011 (HCCH, China). E7 was also used to fabricate the $q$-plates for generating initial vectorial non-separable states. Their molecule structures are illustrated in Fig. S4.

### Sample fabrication
Indium-tin-oxide (ITO) glass substrates (1.5 × 2.0 cm²) were ultrasonically bathed, UV-Ozone cleaned, spin-coated with SD1 solution, and cured at 100 °C for 10 min. For the proposed CLC logic gate, two substrates were separated by 70-μm-thick mylar films and sealed with the epoxy glue to form a cell. The cell gap was checked by Fabry–Pérot interference. After polarized UV exposure (405 nm, 21.78 mW/cm²) for 10 min, the empty cell was imprinted with a uniform alignment perpendicular to the unsealed boundaries. After filled with the CLC material at 80 °C, the cell was gradually cooled to room temperature to form the specific oily streak textures, and kept in the dark for 3 days to achieve the planar textures (Fig. S5). The other thin CLC cell for comparative experiment was formed by 9.5 μm spacers. ITO glass substrates of 3.0 × 3.0 cm² were used to get a larger aperture in Fig. S5. Glass substrates with anti-reflection coating were used in Fig. S7. For the $q$-plate, a 7-μm-thick empty cell was placed at the image plane of a digital-micromirror-device-based dynamic photo-patterning system, and underwent a multistep partly overlapping exposure process[32]. Then, LC E7 was infiltrated into the patterned cell at 80 °C, and gradually cooled to room temperature to get an electrically tunable $q$-plate.

## Numerical simulations

For the input spin cebit $|E_{in}\rangle$ with homogeneous linear polarization, the output state $|E_{out}\rangle$ was calculated by Berreman's 4 × 4 matrix algorithm[37] (Supplementary Note 3) according to the input polarization direction, wavelength, incident angle, and the CLC structure. Thus, all features of the output state could be extracted by standard definitions from its state vector. The transmittance without analyzer, the output polarization direction $\phi_{out}$, the polarization rotation $\Delta\phi$ and the Stokes parameter $S_3$ were calculated, respectively, by

$$\text{Transmittance} = \frac{|\langle E_{out}|E_{out}\rangle|^2}{|\langle E_{in}|E_{in}\rangle|^2}, \tag{4}$$

$$\phi_{out} = \frac{1}{2}\text{Arg}\left(\frac{\langle -\sigma|E_{out}\rangle}{\langle +\sigma|E_{out}\rangle}\right), \tag{5}$$

$$\Delta\phi = \phi_{out} - \phi_{in}, \tag{6}$$

$$S_3 = \frac{|\langle -\sigma|E_{out}\rangle|^2 - |\langle +\sigma|E_{out}\rangle|^2}{|\langle E_{out}|E_{out}\rangle|^2}. \tag{7}$$

Then, for the input vectorial non-separable state, two-dimensional mesh grids were built. The light field within each mesh unit was quasi-uniform, so we were able to employ Berreman's 4 × 4 matrix algorithm to calculate the output field in this unit. Results of all mesh grids composed the complete light field of the output state. The horizontally analyzed intensity profile was given by

$$I_H(x,y) = |\langle H|\psi_{out}\rangle|^2 = \frac{1}{2}|\langle +\sigma|\psi_{out}\rangle + \langle -\sigma|\psi_{out}\rangle|^2. \tag{8}$$

The local polarization ellipses of the output state were also calculated and plotted according to standard formulas.

## Characterization of chiral superstructures

All micrographs of the CLC sample (Fig. 2, S5, S15) were recorded under the reflective mode of an optical microscope (Ci-POL, Nikon, Japan). All macroscopic photographs of the device (Fig. S5, S15) were captured by a camera (EOS M, Canon, Japan). The CLC pitch was measured by Grandjean-Cano wedge method according to the relation of $np/2 = w\tan\gamma$. Here, $w$ is the width across $n$ Cano disclination lines, and $\tan\gamma = 9.15 \times 10^{-3}$ is the tangent value of the wedge angle of the wedge-shape cell (NCLCP, China). As shown in Fig. 2f, $n = 8$, $w = 196.81$ μm, so $p = 450$ nm.

The optical spectra were measured with a halogen light source (iDH2000H-HP, ideaoptics, China) and a spectrometer (PG2000-Pro, ideaoptics, China). For Fig. 2g, S13–S15, LCP/RCP incidence was realized by cascading a wire-grid linear polarizer and an achromatic quarter-wave plate. For Fig. 3, S13, the sample was sandwiched by a fixed polarizer and a rotating analyzer. When the transmission axis of the analyzer was parallel or perpendicular to the output polarization direction, the transmittance would reach the maximum $T_{max}$ or the minimum $T_{min}$, respectively. By combining all these transmittance extremes for various analyzer directions, the output polarization direction $\phi_{out}$ was extracted for each wavelength. The polarization rotation was given by $\Delta\phi = \phi_{out} - \phi_{in}$, and the optical rotatory power was $\Delta\phi / L$, where $L$ was the interaction length. We further calculated the polarization figure of merit (PFM) by[25]

$$\text{PFM} = (T_{max} - T_{min})/(T_{max} + T_{min}). \tag{9}$$

For ideal linear polarizations (i.e., the equal-weighted superimposed states), PFM reaches 1, while for LCP and RCP (i.e., the SAM eigenstates), PFM becomes 0. The relation between PFM and $S_3$ is $\text{PFM} = \sqrt{1 - S_3^2}$.

## Characterization of optical spin cebits

For experimental results in Fig. 4, S12, a linearly polarized beam from a He-Ne laser source (HNL020LB, Thorlabs, USA) was incident at the CLC device mounted on a rotation stage. Here, the surface alignment direction of the device was horizontal, and the rotation axis was vertical. The Stokes parameters of the output homogeneous state were analyzed by a polarimeter (PAX1000VIS, Thorlabs, USA). The input state was also examined with the polarimeter before mounting the CLC device. In this way, we obtained the information of the output spin cebit, such as the polarization direction and the location on the Poincaré sphere.

## Generation and characterization of vectorial non-separable states

The initial vectorial non-separable state was generated by the LC $q$-plate[39]. The $q$-plate is essentially a half-wave plate with inhomogeneous distribution of local optical axes (i.e., LC directors) in the $x$–$y$ plane, formulated as $\alpha(x, y) = q \times \arctan(y/x) + \alpha_0$, where $\alpha$ is the orientation angle of LC director, $\alpha_0$ is a constant, and $q$ is the topological charge of the $q$-plate. The $q$-plate introduces geometric phases (i.e., Pancharatnam–Berry phases) to the input beam with a high conversion efficiency. Ideally, a $q$-plate maps the spin cebit to the vectorial non-separable cebits, which can be expressed as:

$$\mathbf{Q} = \exp(+i2\alpha_0)|+2q\rangle|-\sigma\rangle\langle +\sigma| + \exp(-i2\alpha_0)|-2q\rangle|+\sigma\rangle\langle -\sigma|. \tag{10}$$

During the experiment, the uniform linear polarization state from the He-Ne laser was transformed into the intended VB by the appropriate $q$-plate, namely $q = +0.5$ for Fig. 5, q = −0.5 and $q = +2$ for Fig. S8, respectively. Using a function generator (33500B, Keysight Technologies Inc., USA), a 1 kHz square-wave AC signal at 2.50 V was applied to the LC $q$-plate to maintain the half-wave condition for 632.8 nm.

In Fig. 6, S11, by a 1 × 4 Dammann grating[43], the initial radially polarized VB was diffracted into four equal-energy orders with the same non-separable state but different propagation directions (i.e., the wave vectors). In the case of normal incidence, the grating equation is expressed by $\sin\theta_{phot} = K\lambda/\Lambda_{DG}$, where $K = \pm 1$ and $\pm 3$, $\theta_{phot}$ is the direction of the respective output wave vector, $\lambda = 632.8$ nm, and $\Lambda_{DG} = 46.5$ μm is the grating period. All $\theta_{phot}$ were measured by triangle relation.

The analyzed intensity profiles were captured by a color CCD camera (DCC1645C, Thorlabs, USA) for Fig. 5. To obtain the polarization maps, we measured 6 Stokes intensities with a quarter-wave plate, a polarizer, and a beam profiler (BP 6.4, Femto Easy, France), namely $I_H$, $I_V$, $I_D$, $I_A$, $I_R$, and $I_L$, corresponding to horizontal, vertical, diagonal, anti-diagonal linear polarization, RCP, and LCP, respectively. The azimuth orientation of the polarization ellipse depends on $S_1$ and $S_2$, while the ellipticity depends on $S_3$. For Fig. 5f, a quarter-wave plate and a polarizer were cascaded to examine the circular polarization. Based on the astigmatic transformation, a cylindrical lens ($f = 100$ mm) broke the beam into countable fringes as the criteria of OAM. All beam profiles were observed under the condition that the beam propagates away from the observer.

## Data availability

All data supporting this study and its findings are available within this published article and its supplementary information files. Any other relevant data are available from the corresponding authors upon request.

## Code availability

The algorithms applied in this work can be built following the instructions in Methods and Supplementary Information. The codes that support the findings of this study are also available from the corresponding authors on request.

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

## Acknowledgements

This work was supported by the National Key R&D Program of China (Nos. 2021YFA1202000 (P.C.), and 2022YFA1203700 (S.G.)), the National Natural Science Foundation of China (NSFC) (Nos. 62222507 (P.C.), 12004175 (P.C.), and 62175101 (P.C.)), the Innovation Program for Quantum Science and Technology (No. 2021ZD0301500 (P.C.)), and the Natural Science Foundation of Jiangsu Province (Nos. BK20212004 (Y.L.), and BK20200311 (P.C.)).

## Author contributions

Y.Z., P.C., and Y.L. conceived the original idea. Y.Z. conducted the numerical simulations, fabricated the samples and performed the experiments with the assistance of S.L., P.C., D.Z., and W.C. Y.Z. analyzed the data with the help of P.C., S.L., Z.Z., S.G., Y.W., and Y.L. Y.Z., P.C., and S.L. prepared the initial manuscript. All authors participated in the discussion, and contributed to refining the manuscript. P.C. and Y.L. co-supervised and directed the research.

## Competing interests

The authors declare no competing interests.
