## [Peer Review File · Nature Communications]

Logical rotation of non-separable states via uniformly self-assembled chiral superstructuresREVIEWER COMMENTS

Reviewer #1 (Remarks to the Author):

I have read the article "Logical rotation of non-separable states via uniformly self-assembled chiral superstructures" by Yi-Heng Zhang et al., and I am unfortunately of the opinion that it is not suitable for publication in Nature Communications.

While the manuscript is well written, reports high-quality data complemented by a detailed theoretical analysis and a thorough description of materials/methods, the effect described is not important enough to deserve publication in this journal. The author indeed report of a "logical rotation of non-separable states", which sounds profound, but in my opinion is not. While it is true that the phenomenon observed can be described in terms of a rotation on the high-order Poincaré sphere, and that such rotation resembles logical operations on qubits, I feel that the terminology is here employed inappropriately. A logical control system in its usual meaning is a system where (i) the output state depends on the input state and on a control state following a certain law, (ii) the output of a stage can be inserted into the input and control of a successive stage, such that a logical network can be constructed. As an example, consider the usual electronic Boolean building blocks, that can be assembled all the way up to a complete processor. The system presented in the current manuscript lacks feature (ii), since input/output channels are photonic states, while the control channel is a mechanical rotation of the whole chip. This makes impossible to build a logical network based upon the proposed "logical" element.

I would therefore encourage the authors to present their results in another venue, different from Nature Communications.

Reviewer #2 (Remarks to the Author):

In this manuscript entitled "Logical rotation of non-separable states via uniformly self-assembled chiral superstructures", the authors theoretically revealed and experimentally observed the unusual dependence of the optical rotatory power on the incident angle in the short-wavelength edge vicinity of CLCs. Then, using this controllable logical rotation gate with rotation angle over 4π , a variety of nice and interesting classical counterparts of quantum gates are realized. It presents an elegant strategy to harness high-dimensional photonic states, and provides a compact and practical tool for further exploiting the parallelism between classical non-separability and quantum entanglement. This is an interesting and excellent work of high quality, and I would like to recommend the manuscript for publication after some issues revision.

My comments to the authors:

1. There are some minor aspects in the transmittance spectra that I would suggest to clarify. At line 172-173, the author claim 'the increment of φ_{out} at 632.8 nm reaches 90° with its PFM $> 0.90'$. It might be re-organized to make this point clearer, since 'covers a wide range over 360° ' is also stated in line 191-192.

2. The rotation range of such CLC device is wider than traditional rotators, but I have some concerns on its transmission efficiency. During the execution of logical rotation, oblique incidence is involved, so the reflection from the glass substrates should be considered.
3. The authors design a novel scheme for angular motion tracking. The tilt angle is visualized by the rotation of the intensity profile of vector beams. I suggest to add some discussions on the angle detection range. As I understand, many factors will affect the detection range, such as the working wavelength, the dispersion curve of optical rotatory power, and the interaction length. More details should be added to help readers get a better understanding of its principle and application.
4. Self-organization is a distinctive feature of chiral liquid crystals. In this work, self-assembly is mainly reflected in the easy and low-cost fabrication of 1D chiral photonic crystal. Another important aspect of self-assembled CLCs is their dynamic behavior responsive to external stimuli. For example, with chiral photo-switches introduced, the pitch of helical structures can be modulated by light, and the photonic band can be dynamically tuned. Could this contribute to somehow light-controlled logical rotation of non-separable states?
5. There are some pictures that are not clear enough, like Figure 2h, 3a, 5a, it would be better to represent these pictures with real optical components instead of a simple rectangle.

Reviewer #3 (Remarks to the Author):

In this manuscript, the authors reported a new approach to realizing logical rotation of vectorial non-separable states by self-assembled chiral superstructures. This was experimentally achieved by controlling incident angle of vectorial beams, resulting in polarization rotation at the visible range. Compared to previous works with magneto-optic effect, the major novelty of this work is the wide tuning range over 4π and simple system. Overall, both theoretical calculation and experimental results are elegant and sound. I think the paper deserves publication in Nature Commun., but a few points need to be addressed after revisions.

1. The left-handed superstructure is presented in the manuscript. If right-handed chiral liquid crystal was used, the optical rotatory power would take a minus sign. This indicates an opposite direction of the polarization rotation? I am curious about how the logical rotation angle changes in this opposite chiral structure.
2. The authors claim that the logical rotation angle can be tuned over 4π for 633 nm. In Fig. 2d, the dynamic range of the polarization rotation seems variant for different wavelengths. Is the tuning range of the logical rotation also different? What factors will affect this key parameter of the logical rotation? This needs to be clarified clearly.
3. Very typical examples of non-separable states, such as radially/azimuthally polarized vector beams, are considered in this work. Their logical spaces are special high-order Poincare spheres with $||l|| = 1$. What about other space with a larger l ? Will the proposed chiral liquid crystal logical rotation gate still work?
4. I realize that the chiral liquid crystal is generated by left-handed structures in this work. Is it possible to synthesize right-handed superstructures with right-handed chiral dopant?
5. The authors have demonstrated vector vortex beam with topological charge $l=1$. I suggest the authors to discuss high-order vector vortex, which is also important for high-dimensional photonic states as mentioned in the manuscript.

6. The authors mentioned that the propagation dynamic of light resembles the normal incidence with a longer 'effective wavelength' $\lambda_{\text{eff}} = \lambda / \cos\theta$ for the oblique incidence at an angle θ . Is there any difference for a minus incident angle $-\theta$?

7. The authors mentioned that the chiral superstructures have uniform standing helices. I realize that there are also some defects in the sample. I suggest the authors to discuss more on the reasons and the limits of homogeneous region.

RESPONSE TO THE REVIEWERS:

We greatly appreciate all the reviewers for accessing our manuscript of NCOMMS-23-22930, and providing valuable comments. All reviewers' comments and suggestions are quite helpful for us to substantially improve the quality and readability of this manuscript.

During the past three months, we have devoted much effort to clarifying the proposed concepts by comprehensive theoretical analysis and solid experiments. We have designed a concrete route from the proposed CLC logical rotation gate to a logic network of non-separable states, and performed a set of experimental validations. Besides, all inspiring comments on technical aspects have also been carefully considered. At present, we have thoroughly improved and polished the whole manuscript in a major revision. In particular, we have wholly revised the last section of Results and the Discussion, added two Supplementary Notes, and added ten Supplementary Figures.

In the following, we provide a point-to-point reply to all reviewers' concerns and questions. A list of changes is attached in the end, and all changes are highlighted in blue in the revised manuscript and supplementary information. We hope the new version would be found more convincing and meet the journal's criteria for publication.

We truly appreciate all Reviewers' valuable comments.

Reviewer #1

I have read the article "Logical rotation of non-separable states via uniformly self-assembled chiral superstructures" by Yi-Heng Zhang et al., and I am unfortunately of the opinion that it is not suitable for publication in Nature Communications. While the manuscript is well written, reports high-quality data complemented by a detailed theoretical analysis and a thorough description of materials/methods, the effect described is not important enough to deserve publication in this journal. The author indeed report of a "logical rotation of non-separable states", which sounds profound, but in my opinion is not. While it is true that the phenomenon observed can be described in terms of a rotation on the high-order Poincaré sphere, and that such rotation resembles logical operations on qubits, **I feel that the terminology is here employed inappropriately**. A logical control system in its usual meaning is a system where (i)

the output state depends on the input state and on a control state following a certain law, (ii) the output of a stage can be inserted into the input and control of a successive stage, such that a logical network can be constructed. As an example, consider the usual electronic Boolean building blocks, that can be assembled all the way up to a complete processor. **The system presented in the current manuscript lacks feature (ii), since input/output channels are photonic states, while the control channel is a mechanical rotation of the whole chip.** This makes impossible to build a logical network based upon the proposed "logical" element. I would therefore encourage the authors to present their results in another venue, different from Nature Communications.

Response

We greatly appreciate the respected Reviewer #1 for his or her precious time in reading our manuscript and providing some positive views on some aspects of this work. We fully agree with the insightful summary of two basic requirements of a logic control system. The main concern raised by Reviewer #1 is the gap between (i) the second requirement about inserting the output into the control of the successive stage, and (ii) the operating mode of our currently presented device. In the revision, we devote much effort to building a logic network of non-separable states, including terminology clarification, newly-designed operating mode and network configuration, and comprehensive experiments for verification. For your kind reference, the outline of our revision on this issue is presented graphically in Fig. R1. We try to provide a detailed exposition below, which is organized into 10 sections.

Fig. R1 | Outline of the route to the logic network of non-separable states.

1. Terminology of the logical rotation

Two types of gates are usually applied in quantum network [Quantum computation and quantum information (Cambridge University Press, 2010)]. The first type is the single-qubit gate. The output of this gate is determined by the input following a certain

law. A control is not necessary. Its analogue in Boolean circuit is NOT gate. In quantum circuit, the state space is the Bloch sphere rather than two discrete points, so diverse single-qubit gates can be designed, including Pauli-Z gate, Hadamard gate, S gate, and so on. The second type is the controlled gate, a typical example of which is CNOT gate. The output is determined by both the input qubit and the control qubit, and the output can be further directly inserted into the control of a successive gate. Its analogue in Boolean circuit is two-bit gate, like AND gate or OR gate. Although the first type of gates mainly focus on qubit manipulation, the second-type controlled gates exactly satisfy the two requirements of a logic control system, and can assemble into a complete logic network.

In this study, we propose the concept of ‘logical rotation’ analogous to the qubit rotation. ‘Qubit rotation’ describes the qubit evolution in the form of rotation in the qubit space, which belongs to the first gate type. Here, as a classical counterpart, ‘logical rotation’ is a term to describe the non-separable state evolution in the form of rotation in its logical space. More specifically, compared to the input state, the output state is rotated by an angle Φ around the z -axis in the logical space (i.e., the HOPS), which corresponds to the synchronous rotation of an angle $\Phi/2$ for local linear polarizations in the VB, as displayed in Fig. 1. Thus, ‘logical rotation’ here is essentially a description of the state evolution in the logical space, not necessary to implicate the logic control manner.

Logic control was not a necessity of the proposed logical rotation in our previous manuscript. However, if the logical rotation gate could be upgraded to satisfy the two requirements of the logic control system as mentioned by Reviewer #1, we would anticipate an important step from a single device to the complete logic network of non-separable states. Thus, following Reviewer #1’s valuable suggestions, we make further efforts to exploit the unprecedented capability of the proposed CLC logical rotation gate.

The improvement lies in the control modes. If the logical rotation angle Φ can be regulated by some external factor, a variety of logic gates will be realized with dynamic controllability. This situation has been fully considered previously. Meanwhile, if Φ is controlled by some intrinsic component of the photonic state, the logical rotation gate will be analogous to the CNOT gate. Such control mode would pave the way for a desired logic network of non-separable states.

2. Incident-angle-dependent logical rotation

It is the key principle that the logical rotation of non-separable states depends on the incident angle θ at the CLC device. In fact, the incident angle is determined by both the

beam and the device, formulated as $\theta = \theta_{\text{mech}} - \theta_{\text{phot}}$, where θ_{mech} is the tilt angle of the CLC device and θ_{phot} is the wave vector direction of light. Thus, the logical rotation can be controlled by two modes (Fig. R2). The *mechanical mode* is to keep $\theta_{\text{phot}} = 0^\circ$ and to alter θ_{mech} by mechanically rotating the CLC device, as presented in previous manuscript. While the other mode, named as *wave vector mode*, is to keep θ_{mech} constant and alter the wave vector direction θ_{phot} to achieve controllable logical rotation. Note that θ_{phot} can be altered passively by optical elements like gratings.

Classical bits	Qubits	Logical rotation of non-separable cebits	Features
 NOT One bit	 Qubit rotation Single qubit		 ◆ Mechanical mode ✓ Controlled by mechanics ✓ Angular motion sensor
 AND Two bits	 CNOT Control qubit & target qubit		 ◆ Wave vector mode ✓ Controlled by photonic DoF ✓ Potential for logic network

Fig. R2 | Two control modes of the logical rotation of non-separable states, and their analogues in classical and quantum circuits.

3. *Mechanical mode*: controlled by mechanical rotation

The control mode presented in Fig. 5 is the *mechanical mode*. One non-separable cebit is accepted as an input, and an output cebit is determined by the external physical quantity θ_{mech} . Mechanical rotation of the CLC device is the only required operation to perform a variety of distinguished logic gates. In this mode, the CLC logic gate is analogous to the classical NOT gate and the qubit rotation (including Pauli-Z gate, S gate et al), with an additional parameter θ_{mech} for the controllability (Fig. R2).

Notably, the control is the tilt angle of the device, a mechanical quantity external to the light. The logical rotation controlled by tilt angle θ_{mech} should be understood more precisely as ‘external-parameter dependence’, slightly different from ‘logic control’. Therefore, the first requirement of a logic network is satisfied, but the second is not, as pointed out by the respected Reviewer #1. We cannot build a logic control system only with CLC logical rotation gates in this mechanical mode. This is similar to the fact that a complete logic network cannot be constructed only with single-qubit gates.

4. *Wave vector mode*: controlled by photonic state

In contrast, the control in *wave vector mode* is the photonic wave vector (i.e., θ_{phot}). Distinguished from the mechanical tilt angle θ_{mech} , the wave vector is an intrinsic degree

of freedom (DoF) of light. When the logical rotation is executed, the wave vector DoF acts as the control, and the non-separable state of the spin and the orbital DoFs serve as the target, as illustrated in Fig. R2. Since both the control and the target are carried by the photonic state, the CLC device in *wave vector mode* is similar to two-bit gates for Boolean circuits and CNOT gates for quantum networks (Fig. R2).

Another minor aspect in this mode is worth mentioning. Although the logical rotation is controlled by altering θ_{phot} , the preset constant θ_{mech} can determine the operating region of the CLC device, analogous to the biasing of the electronic transistor to fix the quiescent point at the desired location. Because θ_{mech} is usually much larger than θ_{phot} , a different θ_{mech} would lead to a different operating region, as well as different dependence of the logical rotation angle Φ on θ_{phot} . This also offers a practical way to endow the CLC logic gate with the intended wave-vector-controllability.

5. Experimental verification of the *wave vector mode*

To experimentally demonstrate the logical rotation controlled by photonic wave vector, we design the quantum-like circuit in Fig. R3a and physically implement it with the setup in Fig. R3b. By employing a combination of a q -plate and a 1×4 Dammann grating, we generate the same initial non-separable state with radial polarization, but encoded with different θ_{phot} in four channels.

The 1×4 Dammann grating is a binary-phase (0 and π) grating composed of a set of normalized phase transition points, selected as $\{0, 0.22057, 0.44563, 0.5, 0.72057, 0.94563, 1\}$ [*Appl. Opt.* **34**, 5961-5969 (1995)]. The grating period is $\Lambda_{\text{DG}} = 46.5 \mu\text{m}$. The incident wavelength is $\lambda = 632.8 \text{ nm}$. The normal incident beam is diffracted into four orders with nearly equal-energy distribution, whose directions obey the following grating equations. Channel 1: $\theta_{\text{phot}} = \arcsin(-3\lambda/\Lambda_{\text{DG}})$; Channel 2: $\theta_{\text{phot}} = \arcsin(-\lambda/\Lambda_{\text{DG}})$; Channel 3: $\theta_{\text{phot}} = \arcsin(+\lambda/\Lambda_{\text{DG}})$; Channel 4: $\theta_{\text{phot}} = \arcsin(+3\lambda/\Lambda_{\text{DG}})$. As a result, the VBs in four channels propagate along the direction $\theta_{\text{phot}} = -2.34^\circ, -0.78^\circ, +0.78^\circ, \text{ and } +2.34^\circ$, respectively.

The non-separable states in four channels are simultaneously sent to the same CLC logic gate tilted at $\theta_{\text{mech}} = 17.84^\circ$, and the resultant incident angles are $\theta = \theta_{\text{mech}} - \theta_{\text{phot}} = 20.18^\circ, 18.62^\circ, 17.06^\circ, \text{ and } 15.50^\circ$, respectively. As displayed in Fig. R3c, the analyzed intensity profiles with two separated lobes exhibit different orientations, implying different output states in four channels. Since the input non-separable states are the same, the logical rotation is efficiently controlled by the photonic wave vector. The evolutions of photonic states are vividly presented in Fig. R3d,e. In a word, the

logical rotation can be well controlled in the *wave vector mode*, exhibiting parallel processing manner.

Fig. R3 | **a**, Circuit of the logical rotation controlled by wave vector. The upper horizontal line denotes the wave vector DoF, while the lower line denotes non-separable states. Two short vertical lines with round endpoints denote the control ports of the logical rotation gate. Four channels are processed synchronously for better comparison. **b**, Optical setup for controlling logical rotation by wave vector direction. DG, Dammann grating; LP, linear polarizer; CH, channel. **c**, Measured and simulated results of analyzed intensity profiles at different θ_{phot} . White arrows denote horizontal analyzers. **(d)** Table and **(e)** chart of the photonic state evolution in four channels. Inset: illustrations of the polarization distribution of the non-separable states in step I, and in channel 1, 4 in step III, respectively.

6. Reverse-controlling unit: coordinated with the proposed logical rotation gate

After experimental verification, we now check whether the CLC logical rotation gate in the *wave vector mode* can satisfy the requirements of the logic network. The output non-separable state is determined by the input non-separable state and the wave vector (i.e., θ_{phot}), which meets with the first requirement. The control manner of our CLC device is unidirectional. The wave vector DoF acts as the control, and the non-separable state in the spin and the orbital DoFs serves as the target. On one hand, the CLC logical rotation gate is similar to the CNOT gate on the aspect of controllability (Fig. R2). On the other hand, it is a major difference that the control and the target of a CLC gate belong to distinct DoFs, but those of a CNOT gate are generally the quantum particles of the same kind. Thus, the output of a CNOT gate can directly act as the control of a successive one, while it is indirect for our CLC device.

Fortunately, despite different DoFs, the control and the input/output are all embedded in the photonic state. Accordingly, a reverse-controlling unit is proposed to control the photonic wave vector (i.e., θ_{phot}) based on incident non-separable states (Fig. R1), so that the output of a CLC logical rotation gate can be inserted into the control. By assigning a reverse-controlling unit between two CLC logical rotation gates, the output of the former stage is able to affect the input and the control of the next stage, so the second requirement of logic network is also satisfied.

Generally speaking, a reverse-controlling unit needs to map a set of non-separable states to a set of propagating directions (i.e., photonic wave vector). This task involves sophisticated engineering of the spin, orbital and wave vector DoFs of light. Recent researches in machine-learning inverse-design, meta-atom coupling models and innovative configurations have facilitated diverse forms of light-matter interactions in metasurfaces [*Science* **376**, eabi6860 (2022); *Nature Photonics* **17**, 36-47 (2023)]. Featured by custom-designed manipulation of complex light field, especially those massive VBs [*Adv. Mater.* **32**, 1903983 (2020); *Light Sci. Appl.* **10**, 222 (2021)], metasurfaces are very promising candidates for the reverse-controlling unit.

7. Designed configuration of the logic network of non-separable states

It is noteworthy that the photonic state with multiple DoFs should be understood as a package of multiple components. One component is the non-separable cebit associated with the spin and orbital DoFs, serving as the input and output of the logical rotation. The other component is the photonic wave vector DoF acting as the control. Multiple DoFs in the photonic state allow us to build a logic network of non-separable states using the proposed CLC logical rotation gates coordinated with the reverse-controlling units. The designed configuration is illustrated in Fig. R4a. Non-separable state rotation gates and reverse-controlling units are assembled in an alternative sequence. Controlled by θ_{phot} , the logical rotation gate transforms the input non-separable state into the output state. This output is fed to a reverse-controlling unit, which can modulate θ_{phot} dependent on the non-separable state. In this way, the non-separable state in the former stage is inserted into the wave vector DoF (i.e., θ_{phot}), acting as the control of the successive logical rotation gate. Two requirements of the logic control system are fulfilled simultaneously.

Fig. R4 | **a**, Circuit of a logic network. The upper/lower horizontal line denotes the wave vector DoF or the non-separable state, respectively. Short vertical lines with round endpoints denote the control ports. **b-c**, Circuit (**b**) and optical setup (**c**) of the proof-of-principle logic network. Here, we reduce the non-separable state to the spin cebit (i.e., polarization). The reverse-controlling unit is implemented by the integration of two quarter-wave plates (QWPs) and a polarization gratings (PG) of period $\Lambda_{\text{PG}} = 27.3 \mu\text{m}$. LP, linear polarizer. **d-e**, Measured photonic state evolution table (**d**) and chart (**e**) of the proof-of-principle logic network. The output polarization states in three conditions are illustrated.

8. Logic network of the reduced non-separable states: spin cebits

As proof-of-principle, we design a simplified version of the logic network, so that we can immediately implement it with some easily-accessible optical elements. As shown in Fig. R4b, the basic structure of the logic network remains unchanged, while the treated non-separable state is reduced to the reduced non-separable state with $l = 0$, that is, homogeneous polarization (i.e., spin cebit). Two CLC devices ($\theta_{\text{mech}} = 16.55^\circ$) are applied to clarify how the former stage affects the successive logical rotation. Here, the logical rotation of spin cebit is around the equator of the common Poincaré sphere (Fig. 4), quantified by the polarization orientation ϕ , and the logical rotation angle $\Phi = 2\phi$.

The reverse-controlling unit for spin cebits is physically realized via the polarization grating sandwiched by two quarter-wave plates (QWPs). The polarization grating can be understood as a half-wave plate with linearly varying orientation of local optical axes in the horizontal direction, formulated as $\alpha = -\pi x/\Lambda_{\text{PG}}$. Owing to the geometric phases, the incident beam with LCP/RCP will be deflected to the $-x/+x$ direction, and transformed into the opposite chirality [*J. Lightwave Technol.* **33**, 2068-2077 (2015)]. The QWP before the polarization grating is applied to transform the linearly polarized state into a circularly polarized state, and the QWP after the polarization grating

converts the output beam back to the original linear polarization. The functions of the whole reverse-controlling unit for the spin cebit can be summarized as following. (i) For horizontal polarization input ($\phi = 0^\circ$), θ_{phot} is altered by $\sin\theta_{\text{out}} = \sin\theta_{\text{in}} - \lambda/\Lambda_{\text{PG}}$, and the output remains $\phi = 0^\circ$. (ii) For vertical polarization input ($\phi = 90^\circ$), θ_{phot} is altered by $\sin\theta_{\text{out}} = \sin\theta_{\text{in}} + \lambda/\Lambda_{\text{PG}}$, and the output remains $\phi = 90^\circ$. Clearly, the variation in θ_{phot} is controlled by the incident spin cebit.

9. Experimental verification of the proof-of-principle logic network

The experimental setup is illustrated in Fig. R4c. The operating wavelength is $\lambda = 632.8$ nm. The period of the polarization grating is $\Lambda_{\text{PG}} = 27.3$ μm . In this network, all CLC logic gates are kept static without any mechanical rotation. Notably, we only prepare the spin cebit and wave vector of the initial photonic state (step I), and all subsequent states (step II, III, IV) evolve automatically by the aforementioned control rules. Three typical conditions of the initial state are performed, and the polarization states in the network are measured with polarimeter and presented in Fig. R4d,e.

Condition 1 and Condition 2 share the same initial spin cebit $\phi = 0.0^\circ$ (step I). Owing to different wave vector directions, the output of the first CLC logical rotation gate (step II) is $\phi = 0.2^\circ$ for Condition 1 and $\phi = 89.4^\circ$ for Condition 2, respectively. The different output spin cebit leads to negative/positive variation in θ_{phot} via the reverse-controlling unit (from step II to step III). Thus, the control of the second logical rotation gate (step III) is modulated in response to the output of the first gate (step II).

Comparison of Condition 2 and Condition 3 will further explain the function of the reverse-controlling unit. In the output of the first CLC logical rotation gate (step II), both photonic states share the same wave vector direction $\theta_{\text{phot}} = -2.55^\circ$, while their spin cebit are $\phi = 89.4^\circ$ and $\phi = 0.0^\circ$, respectively. Dependent on these spin cebits in step II, the reverse-controlling unit alters the wave vector and mainly keeps respective spin cebit, producing the state $\theta_{\text{phot}} = -1.22^\circ$, $\phi = 89.9^\circ$ and the state $\theta_{\text{phot}} = -3.88^\circ$, $\phi = 0.4^\circ$ in step III, respectively. After the second CLC logical rotation (step IV), the output spin cebit becomes $\phi = 148.2^\circ$ and $\phi = 119.9^\circ$, respectively. Corresponding polarization rotation from step III to IV for Condition 2 is $\Delta\phi = 148.2^\circ - 89.9^\circ = 58.3^\circ$ (modulo 180°), corresponding to logical rotation angle $\Phi = 0.65\pi$, which shows clear contrast to $\Delta\phi = 119.9^\circ - 0.4^\circ = 119.5^\circ$ (modulo 180°) and $\Phi = 1.33\pi$ for Condition 3.

In this proof-of-principle network, (i) the output spin cebit of the CLC logical rotation gate is determined by the input spin cebit and the wave vector DoF, and (ii) the output spin cebit of the former stage is inserted into the wave vector DoF via a reverse-

controlling unit, acting as the control of a successive stage. Two requirements of the logic network are fulfilled by our proposed configuration.

10. Conclusion

To build a logic network of non-separable states, the logical rotation needs to be controlled by a photonic state, and the output non-separable state of the former stage needs to be inserted into the control and the input of the successive logical rotation gate. To satisfy these two requirements, on one hand, we have introduced the photonic wave vector direction (θ_{phot}), an intrinsic DoF of light, as another control method of the logical rotation (Fig. R2), distinguished to the mechanical rotation mode reported in previous manuscript. The logical rotation controlled by the photonic wave vector is well validated by experiments (Fig. R3). On the other hand, a reverse-controlling unit is proposed to modulate θ_{phot} based on the non-separable state. CLC logical rotation gates and reverse-controlling units assemble sequentially into a logic network (Fig. R4).

In the proof-of-principle demonstration (Fig. R4), we consider the reduced non-separable states with $l = 0$, i.e., spin cebits, so that a simplified logic network can be experimentally demonstrated with easily-accessible optical elements. For more general non-separable states, versatile metasurfaces capable of manipulating multiple VBs are promising candidates for the reverse-controlling unit. A step from the proposed single logical rotation gate towards the logic network is vital for developing algorithms or protocols based on non-separable states. The composed functional logic network would have great potentials in high-dimensional photonics and optical informatics.

IN ALL, we greatly appreciate the respected Reviewer #1's valuable suggestion. In this revision, we have revised the terminology clarification, proposed newly-designed operating mode and network configuration, and conducted comprehensive experiments for the verification. The manuscript has been wholly revised, especially in Principles and the last section of Results. Besides, the discussions on the logical rotation controlled by photonic wave vector have been added as Supplementary Note 5, and the discussions on the logic network and its demo for spin cebits have been added as Supplementary Note 6. We have also added more results in Figure 6, and Supplementary Fig. S10-S12. We really hope the new version would be found more convincing, and suitable for publication now.

Reviewer #2

In this manuscript entitled “Logical rotation of non-separable states via uniformly self-assembled chiral superstructures”, the authors theoretically revealed and experimentally observed the unusual dependence of the optical rotatory power on the incident angle in the short-wavelength edge vicinity of CLCs. Then, using this controllable logical rotation gate with rotation angle over 4π , a variety of nice and interesting classical counterparts of quantum gates are realized. It presents an elegant strategy to harness high-dimensional photonic states, and provides a compact and practical tool for further exploiting the parallelism between classical non-separability and quantum entanglement. **This is an interesting and excellent work of high quality, and I would like to recommend the manuscript for publication after some issues revision.**

Response

We greatly appreciate the respected Reviewer #2 for his or her very positive comments on the novelty and significance of our work, and kind recommendation for publication. According to the following valuable suggestions, we have substantially improved the quality of our manuscript. We hope the new version would be found more convincing and suitable for publication now.

My comments to the authors:

1. There are some minor aspects in the transmittance spectra that I would suggest to clarify. At line 172-173, the author claim ‘the increment of ϕ_{out} at 632.8 nm reaches 90° with its PFM > 0.90 ’. It might be re-organized to make this point clearer, since ‘covers a wide range over 360° ’ is also stated in line 191-192.

Response

We are sorry for this confusion. Following Reviewer #2’s kind reminding, we have added some indicator lines in the transmittance spectra. Here, as shown in Fig. R5, the nearly 90° increment of ϕ_{out} is under the condition that the incident angle θ changes from 0° to 14° . These are two exemplary cases, but not the ultimate limit of our CLC logical rotation gate. When θ ranges from 0° to 26° , ϕ_{out} can be tuned over 360° , as shown in Fig. 4. Figure R5 has been added as the revised Fig. 3c, and related description has been revised in Line 179-181 in the revised manuscript.

Fig. R5 | Spectrally resolved output polarization direction ϕ_{out} (red scatters) and PFM (gray curves) at $\theta = 0^\circ$ and 14° , respectively. As marked by horizontal and vertical red drop lines, the increment of ϕ_{out} approaches 90° at 632.8 nm.

2. The rotation range of such CLC device is wider than traditional rotators, but I have some concerns on its transmission efficiency. During the execution of logical rotation, oblique incidence is involved, so the reflection from the glass substrates should be considered.

Response

We totally agree with Reviewer #2’s thoughtful comments. In our uniformly self-assembled chiral superstructures, the loss in transmission efficiency is mainly attributed to the reflection from the substrates. To examine the values of the transmission efficiency, we used two CLC devices made of common ITO substrates and anti-reflection substrates, respectively. Under 632.8 nm incidence, the transmittances at various angles are measured by a photodiode power sensor. As displayed in Fig. R6, the average transmission efficiency of the presented device is 79.58%, and can be further improved to 93.58% with anti-reflection substrates. Accordingly, Fig. R6 has been added as Fig. S7, and corresponding discussions have been added in Line 224-225 in the revision.

Fig. R6 | Transmittance of the CLC device under 632.8 nm laser illumination at different incident angles. The substrates are (a) common indium-tin-oxide (ITO) glass (blue), and (b) glass with anti-reflection coating (red), respectively.

3. The authors design a novel scheme for angular motion tracking. The tilt angle is visualized by the rotation of the intensity profile of vector beams. I suggest to add some discussions on the angle detection range. As I understand, many factors will affect the detection range, such as the working wavelength, the dispersion curve of optical rotatory power, and the interaction length. More details should be added to help readers get a better understanding of its principle and application.

Response

We thank Reviewer #2 for the positive view on the scheme for angular motion tracking, and the valuable suggestions on clarifying the key factors. The principle of the proposed angle sensor and those key factors will be discussed in the following. A series of comparative experiments have also been performed during the revision.

Principles. The controllable logical rotation of non-separable states is established on the incident-angle-dependent optical rotatory power of CLC structures. When the operating wavelength is located in the short-wavelength edge vicinity of the photonic band gap, the rotation of polarization is sensitive to the incident angle. In other word, a connection exists between the non-separable state and the tilt angle of the CLC device, θ_{mech} . As shown in Fig. 5b-e, using a horizontal analyzer, the output non-separable state is characterized by the orientation angle β of the dark fringe (i.e., the intensity minima) segmenting the analyzed intensity profile. β is extracted via image processing (Fig. S9). By setting a series of θ_{mech} and measuring corresponding β , a working curve (i.e., β - θ_{mech} relation) of the CLC angle sensor is built (Fig. 6b), and can be applied for angular motion tracking. Through collecting the analyzed intensity profiles of the output states in sequence, and referring to the working curve, we are able to obtain the angular motion of the device.

The first key factor: L . For longer interaction length L , the dependence of the polarization rotation on the incident angle becomes stronger (quantitatively speaking, $\Delta\phi/\Delta\theta$ is larger). The polarization rotation originates from the phase retardation between two eigenmodes during the propagation through the CLCs. Thus, on one hand, sufficient L is essential for the accumulation of enough phase retardation, assuring recognizable polarization difference in the output state. In contrast, we fabricate a CLC device with the same material but a much thinner cell gap, $L^* = 9.5 \mu\text{m}$. In this case, $L^* \approx 21p$, much lower than the presented $L \approx 155p$ by an order of magnitude. The optical properties of this device are presented in Fig. R7. Compared to Fig. 3b, only

few transmittance extremes can be observed in Fig.R7e, which implies that the polarization rotation range is quite limited. On the other hand, though extremely long interaction length could greatly enhance the polarization rotation, it becomes more challenging to fabricate the uniform chiral superstructures for ultra-long standing helices.

Fig. R7 | Optical properties of the CLC logic gate with **short interaction length** ($L^* = 9.5 \mu\text{m}$). **a-c**, Simulated results of **(a)** the transmittance without analyzer, **(b)** polarization rotation $\Delta\phi$, and **(c)** S_3 as a function of θ and the input polarization direction ϕ_{in} , for the operating wavelength 632.8 nm. **d**, Measured transmittance spectra for normal incidence with RCP (purple) and LCP (yellow), respectively. **e**, Measured co-polarized (red) and cross-polarized (blue) transmittance spectrum for normal incidence with $\phi_{\text{in}} = 90^\circ$.

The second key factor: $\lambda/(n_0p)$, namely the ratio of operating wavelength to the short-wavelength edge of PBG. When $\lambda/(n_0p)$ increases, λ gets closer to the PBG and the dispersion of optical rotatory power becomes stronger, so the logical rotation angle Φ depends more sensitively on the incident angle θ , while their dynamic ranges are also affected. To explain this more vividly, a comparative experiment has been performed. By increasing the chiral dopant ratio to 2.28 wt%, we make a left-handed CLC device with shorter p , so λ is closer to the short-wavelength edge ($\lambda = 632.8 \text{ nm}$ and $n_0p = 670 \text{ nm}$). The simulated results in Fig. R8a indicate a stronger dependence of $\Delta\phi$ on θ . When $\theta = 4.0^\circ, 8.9^\circ, 11.0^\circ$, and 14.8° , the CLC device performs an identity operator, a T gate, an S gate, and a Pauli-Z gate, respectively (Fig. R8c). Comparing to the results in Fig. 5,6, the device here presents higher sensitivity, whether as a logical rotation gate or an angular motion sensor. Meanwhile, as shown in Fig. R8b,c, $\text{PFM} < 0.85$ for $\theta > 23.0^\circ$, and the PBG already covers λ when $\theta = 29.0^\circ$, indicating a narrower dynamic range of

θ . In a word, in order to achieve desired sensitivity or dynamic range for logical rotation and angle sensing, we should make overall consideration of the operating wavelength and the PBG of the fabricated chiral superstructures.

Fig. R8 | Optical properties of the CLC logic gate when the operating wavelength is closer to the short-wavelength edge of PBG ($\lambda = 632.8$ nm and $n_{op} = 670$ nm). **a**, Simulated dependences of $\Delta\phi$ on the wavelength and θ for $\phi_{in} = 0^\circ$. **b**, Measured transmittance spectra of the CLC device under RCP (purple) and LCP (yellow) incidence at various incident angles θ . **c**, Captured analyzed intensity profiles, and simulated polarization maps of the output state.

Other factors. Chirality of CLC is also an important factor. The polarization rotation, as well as the logical rotation, originates from the phase retardation between two eigenmodes in CLC. In left-handed CLC, the slower mode is right-handed circular polarized (RCP) and the faster mode is left-handed circular polarized (LCP). While in right-handed CLC, the slower mode is LCP and the faster mode is RCP. Thus, a CLC device with opposite chirality leads to an opposite direction of the logical rotation. Despite opposite phenomenon, both left- and right-handed CLC device can indicate the angular motion.

Practice. In the light of above principles, we can design a competent sensor for angular motion tracking by properly setting these key factors. For example, if high sensitivity is desired, L should be long enough, and $\lambda/(n_{op})$ should be large by using appropriate ratio of chiral dopants. If wide dynamic range is preferred, $\lambda/(n_{op})$ should be small but still within the short-wavelength edge vicinity, so that the short-wavelength

edge of PBG will be relatively away from the operating wavelength and the sensor can work effectively until a large tilt angle.

As suggested by Reviewer #2, above discussions have been added in Line 251-256, and Line 296-306 in the revised manuscript. Figure R7,R8 have also been added as Fig. S13,S14, respectively.

4. Self-organization is a distinctive feature of chiral liquid crystals. In this work, self-assembly is mainly reflected in the easy and low-cost fabrication of 1D chiral photonic crystal. Another important aspect of self-assembled CLCs is their dynamic behavior responsive to external stimuli. For example, with chiral photo-switches introduced, the pitch of helical structures can be modulated by light, and the photonic band can be dynamically tuned. Could this contribute to somehow light-controlled logical rotation of non-separable states?

Response

We totally agree with Reviewer #2's insightful views on self-assembled chiral superstructures, and greatly appreciate the innovative idea of the light-controlled logical rotation. To verify its feasibility, we have conducted more theoretical analysis on this issue. As shown in Fig. R9a, under normal incidence, when the variable helical pitch p^* becomes shorter, the PBG from $n_o p$ to $n_e p$ is blue shifted, so the operating wavelength is closer to the PBG, and the optical rotatory power grows accordingly. Following the theory in Supplementary Note 2, the dependence of optical rotatory power on the pitch is calculated and presented in Fig. R9b, which is comparable to that of the proposed oblique incidence.

Furthermore, we consider a radially polarized VB as an input, and simulated the output non-separable states from the chiral superstructures with different p^* . As vividly exhibited in Fig. R9c, the CLC device performs an identity operator, a T gate, an S gate, and a Pauli-Z gate when $p^* = 444.1$ nm, 442.8 nm, 441.3 nm, and 439.1 nm, respectively. With proper active CLC materials, dynamic pitch responsive to external-stimuli is achievable, contributing to controllable logical rotation of non-separable states. For example, chiral photo-switches would enable tunable pitch via light irradiation, and thermoresponsive chiral dopants would enable temperature-dependent pitch.

In a word, this innovative strategy of controllable logical rotation based on dynamically self-assembled superstructures can be rationally expected, and may facilitate multiple external-field-controlled manipulation of high-dimensional photonic states. Corresponding discussion has been added in the section of Discussion, and Fig. R9 has been added as Fig. S16.

Fig. R9 | **a**, Analytical dependence of the PBG on the variable helical pitch p^* under normal incidence. The original pitch $p = 450$ nm. $\lambda = 632.8$ nm. **b**, Analytical optical rotatory power as a function of the incident angle θ (blue), and the tunable pitch p^* (red), respectively. **c**, Simulated analyzed intensity profiles and polarization maps of the output state with different p^* . The input state is radially polarized. White arrows denote horizontal analyzers.

5. There are some pictures that are not clear enough, like Figure 2h, 3a, 5a, it would be better to represent these pictures with real optical components instead of a simple rectangle.

Response

We appreciate Reviewer #2's kind advice. The optical components in the revised Fig. 2,3,5,6 and the newly added Fig. S12 have been represented in the realistic style to give a more vivid illustration.

Note: Please refer to the response of Reviewer #3's comments in the NEXT PAGE.

Reviewer #3

In this manuscript, the authors reported a new approach to realizing logical rotation of vectorial non-separable states by self-assembled chiral superstructures. This was experimentally achieved by controlling incident angle of vectorial beams, resulting in polarization rotation at the visible range. Compared to previous works with magneto-optic effect, the major novelty of this work is the wide tuning range over 4π and simple system. **Overall, both theoretical calculation and experimental results are elegant and sound. I think the paper deserves publication in Nature Commun., but a few points need to be addressed after revisions.**

Response

We truly appreciate the respected Reviewer #3 for his or her very positive comments and kind recommendation for publication. We have carefully revised our manuscript according to Reviewer #3's constructive comments, which are quite helpful for substantially improving the quality and readability of our manuscript. We hope the new version would be found more convincing and suitable for publication now.

1. The left-handed superstructure is presented in the manuscript. If right-handed chiral liquid crystal was used, the optical rotatory power would take a minus sign. This indicates an opposite direction of the polarization rotation? I am curious about how the logical rotation angle changes in this opposite chiral structure.

Response

The understanding of Reviewer #3 is exactly correct. The polarization rotation, as well as the logical rotation, originates from the phase retardation between two eigenmodes in CLC. In left-handed CLC, the slower mode is RCP and the faster mode is LCP. While in right-handed CLC, the slower mode is LCP and the faster mode is RCP. Thus, a CLC device with opposite chirality leads to an opposite direction of the logical rotation.

For clear illustration, we have further performed analytical calculation, numerical simulation and direct experiments. As indicated by Fig. R10a, the optical rotatory power takes a minus sign for right-handed CLCs. In the short-wavelength edge vicinity, the optical rotatory power becomes more negative when the wavelength gets closer to the PBG. Numerical results in Fig. R10b show that the polarization rotation $\Delta\phi$ varies from 180° down to 0° when the incident angle θ increases, whose direction is opposite to that of the left-handed chiral structure. A right-handed CLC device (Fig. R10d,e) is fabricated for experimental verification of the logical rotation. As shown in Fig. R10f, when $\theta = 14.0^\circ, 15.5^\circ, 17.0^\circ, \text{ and } 19.1^\circ$, $\Phi = 0, -\pi/4, -\pi/2, \text{ and } -\pi$, respectively. Visually,

the local polarizations in VBs rotate simultaneously in the negative direction. It is clear that a negative value of the logical rotation angle is introduced in this opposite chiral structure.

Related discussion has been added in the revised Supplementary Note 4 and the revised section of Discussion. Figure R10 has been added as Fig. S15 to give a comprehensive clarification.

Fig. R10 | Optical properties of the CLC logic gate of the opposite chirality (right-handed). **a**, Analytical optical rotatory power (ORP) and polarization figure of merit (PFM) as functions of the wavelength for linearly polarized normal incidence. Inset: the input (ϕ_{in}) and output (ϕ_{out}) linear polarization direction, and the polarization rotation ($\Delta\phi$). k denotes the wave vector. **b-c**, Simulated dependences of **(b)** the polarization rotation $\Delta\phi$, and **(c)** the Stokes parameter S_3 on the incident angle θ and the input polarization direction ϕ_{in} . **d**, Schematic of the right-handed chiral superstructures of pitch p , the reflective optical micrograph of the region indicated by a black dotted circle, and the macroscopic photograph of the CLC device. The scale bars are 100 μm and 1 cm, respectively. **e**, Measured transmittance spectra for normal incidence with RCP (purple) and LCP (yellow), respectively. **f**, Captured analyzed intensity profiles, and simulated polarization maps of the output state.

2. The authors claim that the logical rotation angle can be tuned over 4π for 633 nm. In Fig. 2d, the dynamic range of the polarization rotation seems variant for different wavelengths. Is the tuning range of the logical rotation also different? What factors will affect this key parameter of the logical rotation? This needs to be clarified clearly.

Response

Following Reviewer #3's valuable suggestion, we have further designed and performed a series of comparative experiments to better clarify which factors will affect the dynamic range of the logical rotation.

To study one key factor, the interaction length L , we fabricate a CLC device with the same material but a much thinner cell gap, $L^* = 9.5 \mu\text{m}$. In this case, $L^* \approx 21p$, much lower than the presented $L \approx 155p$ by an order of magnitude. The optical properties of this thin device are presented in Fig. R11. As implied by Fig. R11b, the total dynamic range of the polarization rotation $\Delta\phi$ is less than 90° , corresponding to a range of logical rotation less than π . Furthermore, compared to that of Fig. 3b, only few transmittance extremes can be observed in Fig. R11e, which also implies that the polarization rotation range is quite limited. Thus, on one hand, sufficient L is essential for recognizable polarization difference in the output state when altering the incident angle. For longer interaction L , $\Delta\phi/\Delta\theta$ is larger, so the tuning range of polarization rotation and logical rotation becomes wider for a fixed range of θ . On the other hand, though extremely long interaction length could drastically extend the tuning range, it is very challenging to fabricate the uniform chiral superstructures with ultra-long standing helices.

Fig. R11 | Optical properties of the CLC logic gate with short interaction length ($L^* = 9.5 \mu\text{m}$). **a-c**, Simulated results of **(a)** the transmittance without analyzer, **(b)** polarization rotation $\Delta\phi$, and **(c)** S_3 as a function of θ and the input polarization direction ϕ_{in} , for the operating wavelength 632.8 nm . **d**, Measured transmittance spectra for normal incidence with RCP (purple) and LCP (yellow), respectively. **e**, Measured co-polarized (red) and cross-polarized (blue) transmittance spectrum for normal incidence with $\phi_{\text{in}} = 90^\circ$.

Another key factor is the ratio of operating wavelength to the short-wavelength edge of PBG (i.e., λ/n_{op}). To study this influence, we fabricate a CLC structure with shorter

pitch by increasing the chiral dopant ratio to 2.28 wt% ($\lambda = 632.8$ nm and $n_{op} = 670$ nm). The simulated result in Fig. R12a indicates stronger dependence of $\Delta\phi$ on θ . When $\theta = 4.0^\circ, 8.9^\circ, 11.0^\circ,$ and 14.8° , the CLC device experimentally performs an identity operator, a T gate, an S gate, and a Pauli-Z gate, respectively (Fig. R12c). Comparing to the results in Fig. 5,6, the CLC logical rotation gate here presents higher sensitivity. When $\lambda/(n_{op})$ increases, λ gets closer to the PBG and the dispersion of optical rotatory power becomes stronger, so the logical rotation angle Φ depends more sensitively on θ . As a result, for a given range of θ , the logical rotation angle can cover a wider range. Meanwhile, the tuning range is also affected. As shown in Fig. R12b,c, $\text{PFM} < 0.85$ for $\theta > 23.0^\circ$, and the PBG already covers λ when $\theta = 29.0^\circ$. Because λ is closer to the PBG, λ will be included in the PBG at a smaller θ , which means that only in a narrower θ range can the CLC device act as a competent logical rotation gate. In a word, in order to achieve desired tuning range for the logical rotation, we should make overall consideration of the operating wavelength and the PBG of the fabricated chiral superstructures.

We have added above contents about the logical rotation range in the section of Discussion in the revised manuscript, and added Fig. R11,R12 as Fig. S13,S14, respectively.

Fig. R12 | Optical properties of the CLC logic gate when the operating wavelength is closer to the short-wavelength edge of PBG ($\lambda = 632.8$ nm and $n_{op} = 670$ nm). **a**, Simulated dependences of $\Delta\phi$ on the wavelength and θ for $\phi_{in} = 0^\circ$. **b**, Measured transmittance spectra of the CLC device under RCP (purple) and LCP (yellow) incidence at various incident angles θ . **c**, Captured analyzed intensity profiles, and simulated polarization maps of the output state.

3. Very typical examples of non-separable states, such as radially/azimuthally polarized vector beams, are considered in this work. Their logical spaces are special high-order Poincare spheres with $|l| = 1$. What about other space with a larger l ? Will the proposed chiral liquid crystal logical rotation gate still work?

Response

Thanks for this valuable suggestion. First of all, we would explain the general applicability of our proposed CLC logic rotation gate by theoretical analysis. Logical rotation is the state evolution around the z-axis of corresponding HOPS, which can be executed by introducing certain relative phase shift between two basis states of non-separable states. The propagation through the uniform CLC induces the phase retardation between polarization states, and this manipulation on the spin DoF will be coupled into the orbital DoF due to their intrinsic non-separability, which leads to the logical rotation of the non-separable state. The certain value of l is a component of the basis states, but not involved in this relative phase shift associated with the logical rotation. Thus, the same logical rotation trajectory can be obtained regardless of l .

In the schematic illustration of three representative HOPSs with $l = +1$, $l = -1$, and $l = +4$ (Fig. R13), the non-separable states at representative points on the equator of HOPS are depicted by polarization maps. The vectorial non-separable state is featured by the donut-shaped intensity profile of radius proportional to $\sqrt{|l|}$, and its local polarization orientation rotates $|2\pi l|$ per circulation about the central singularity. As the state logically rotates around the equator, the local polarization directions of the VB rotate simultaneously (see each row in Fig. R13). The relative rotation angle of local polarization only depends on the evolution on HOPS (i.e., $\Delta\phi = \Phi/2$), and does not vary for different l .

Fig. R13 | Schematic illustration of the logical space of vectorial non-separable states, namely the HOPS, spanned by the basis vectors of $|-\sigma\rangle|+l\rangle$ and $|+\sigma\rangle|-l\rangle$, with (a) $l = +1$, (b) $l = -1$, and (c) $l = +4$, respectively. (Θ, Φ) is the spherical coordinate. For certain points at the equator, polarization maps of the non-separable states are depicted.

Additional experiments have also been performed to verify the logical rotation of non-separable states in different logical spaces (Fig. R14). When the incident angle increases, the local polarization is rotated to the positive direction in the VB with either $l = -1$ or $l = +4$, as vividly shown by the simulated polarization maps. While, the analyzed intensity profile with $|2l|$ separated lobes changes in different way, because of different global polarization distributions. When $\theta = 14.0^\circ, 15.5^\circ, 17.0^\circ$, and 19.1° , the CLC device performs an identity operator, a T gate, an S gate, and a Pauli-Z gate. A similar evolution trajectory is obtained on respective HOPS, regardless of l , which agrees well with the theoretical prediction.

In all, the proposed CLC logical rotation gate is applicable for other non-separable states with larger l . Corresponding discussion has been added in Line 226-227 in the revised manuscript, and the revised Supplementary Note 1. Figure R13,R14 have been added as Fig. S1,S8, respectively.

Fig. R14 | Logical rotation of the vectorial non-separable states at 632.8 nm with (a) $l = -1$ and (b) $l = +4$, respectively. Schematic illustrations of the state on HOPS, captured analyzed intensity profiles, and simulated polarization maps of the initial state, and the output state.

4.I realize that the chiral liquid crystal is generated by left-handed structures in this work. Is it possible to synthesize right-handed superstructures with right-handed chiral dopant?

Response

We can fabricate the right-handed chiral superstructures by mixing the nematic LC E7 with right-handed chiral dopant R5011. The empty cell was firstly imprinted with a uniform alignment perpendicular to the unsealed boundaries. After filled with the right-handed CLC material at 80 °C, the cell was gradually cooled to room temperature to form the specific oily streak textures, and kept in the dark for 3 days to achieve the desired planar textures. A right-handed CLC logic gate was successfully prepared (Fig. R15a). The photonic band gap shows the selective reflection of RCP (Fig. R15b), consistent with its handedness.

Fig. R15 | **a**, Schematic of the right-handed chiral superstructures of pitch p , the reflective optical micrograph of the region indicated by a black dotted circle, and the macroscopic photograph of the CLC device. The scale bars are 100 μm and 1 cm, respectively. **b**, Measured transmittance spectra for normal incidence with RCP (purple) and LCP (yellow), respectively.

5. The authors have demonstrated vector vortex beam with topological charge $l=1$. I suggest the authors to discuss high-order vector vortex, which is also important for high-dimensional photonic states as mentioned in the manuscript.

Response

The proposed CLC device is able to execute the logical rotation of higher order non-separable states. Firstly, the logical rotation of non-separable states is based on the synchronous rotation of the spatially-distributed local polarizations. The polarization rotation originates from the incident-angle-dependent optical rotatory power in CLCs. Thus, the logical rotation of the non-separable state is not affected by its topological charge. Secondly, though the beam diameter of a higher order VB is larger, our proposed CLC device has a large enough region to work properly. More detailed discussions and additional experimental results are also involved in the response to the 3rd comment and Fig. R14.

6. The authors mentioned that the propagation dynamic of light resembles the normal incidence with a longer ‘effective wavelength’ $\lambda_{\text{eff}} = \lambda / \cos\theta$ for the oblique incidence at an angle θ . Is there any difference for a minus incident angle $-\theta$?

Response

Thanks for this insightful question. The physical problem here is the polarization state evolution in the chiral anisotropic optical media at oblique incidence. We can conclude that $-\theta$ is equivalent to $+\theta$, after considering the system symmetry. The incident light stimulates certain composition of eigenmodes in CLC based on mode matching conditions at the front boundary. The eigenmodes propagate through the chiral media and produce the transmitted light at the rear boundary. The mode matching conditions at the boundary include the continuity of tangential components of the wave vector and the optical polarization. Thus, we need to consider the incident wave vector, the incident polarization and the chiral superstructures. Firstly, the incident angle changing from $+\theta$ to $-\theta$ means the wave vector is rotated by 180° around the helical axis of the chiral superstructures. Secondly, the chiral superstructures have the symmetry in 180° rotation around the helical axis. Thirdly, the polarization is attached to the wave vector. Thus, the boundary conditions at $+\theta$ and $-\theta$ are identical, and the resultant polarization rotations are the same. Numerical results in Fig. R16 also provide a vivid proof of above claims. Related descriptions have been added in the section of Discussion in the revised manuscript.

Fig. R16 | Simulated results of the polarization rotation $\Delta\phi$ and S_3 as a function of the input polarization direction ϕ_{in} and incident angle $+\theta/-\theta$ (left/right column) for the operating wavelength of 632.8 nm.

7. The authors mentioned that the chiral superstructures have uniform standing helices. I realize that there are also some defects in the sample. I suggest the authors to discuss more on the reasons and the limits of homogeneous region.

Response

We greatly appreciate Reviewer #3's constructive suggestion. The defects are attributed to the self-assemble process of CLC materials. Just after the CLC mixture is cooled from the isotropic phase to the ordered LC phase, non-standing helices appear and gather to form line defects of various widths, taking on the appearance of oily streaks. Owing to the system's tendency to minimize the free energy, the oily streaks gradually merge and transform into uniform standing helices guided by the surface alignment layer. Cell gap is an important factor in this process. In traditional CLC devices, the cell gap is usually less than 10 μm , so the surface anchoring is strong enough to realize uniform planar textures in very short time. In our proposed device with a cell gap of 70 μm , the CLC bulk scales up, so the relative effect of surface anchoring is weakened, and the self-organization takes a longer time of several days. If the cell gap grows to hundreds of micrometers, surface alignment is not enough to realize uniform structure, and external electricity field can be applied to control the alignment of cholesteric helices in the entire bulk [Appl. Phys. Rev. **10**, 011413 (2023)].

Fig. R17 | Fabrication and photographs of the uniform chiral superstructures. **a-d**, Reflective polarized micrographs of the CLC logic gate (**a**) at 80 °C, (**b**) cooled to room temperature, and (**c**, **d**) after kept in the dark for 3 days, respectively. **e-g**, Macroscopic photographs of the CLC device sandwiched by (**e**) crossed polarizers, (**f**) parallel polarizers, and (**g**) no polarizer, respectively. **h-j**, Macroscopic photograph (**h**) and reflective polarized micrographs (**i**, **j**) of a large aperture CLC device with the functional region of 5.8 cm². The scale bars in micrographs and macroscopic photographs are 100 μm and 1 cm, respectively.

As shown in Fig. R17, most of the oily streaks have successfully transformed into planar textures of uniform standing helices, which are optically transparent and clear. The remaining defects gather in the narrow ribbon-like region (Fig. R17e-g). Very limited defect lines are sparsely distributed in the broad planar texture region, without

obvious influence on its performance of the logical rotation. The total functional region is over 1 cm^2 (Fig. R17g) and large enough to manipulate non-separable states, since the treated VBs in our experiments are only about 1 mm in beam diameter.

In fact, the aperture of the CLC device can be further scaled up by using the substrates of sufficient size. As an example, we fabricate a CLC logical rotation gate with $3 \times 3 \text{ cm}^2$ substrates and achieve a functional region of about 5.8 cm^2 (Fig. R17h). This proves the scalability of the proposed self-assemble fabrication technique. Corresponding discussions have been added in Line 165-166 in the revised manuscript, and Fig. R17 has been added as the new Fig. S5.

LIST OF CHANGES

- (1) Some introductory descriptions about the proposed logical rotation gate have been revised. **Line 22-24** in **Abstract** has been revised as ‘The logical rotation angle of a non-separable state can be tuned in a wide range over 4π by this single homogeneous device, flexibly providing a set of distinguished logic gates. Potential applications, including angular motion tracking and proof-of-principle logic network, are demonstrated by specific configuration’. **Line 67-69** in the section of **Introduction** has been revised as ‘Moreover, our proposed device enables angular motion tracking, and serves as a potential building block for the logic network of non-separable states. This study demonstrates a practical scheme to tailor non-separable states in a dynamic and multi-functional way’.
- (2) The terminology of the logical rotation has been revised as ‘Inspired by the qubit rotation, we focus on the logical rotation of such classical non-separable state around the z-axis of its logical space, namely the axis perpendicular to the equatorial plane of the HOPS. Such logical rotation operator can be expressed as the tensor product form’ in **Line 91-93**, and ‘The output state is rotated by an angle Φ around the z-axis of the same HOPS (namely its logical space), which corresponds to the synchronous rotation of an angle $\Phi/2$ for local linear polarization of the input VB. If Φ can be regulated by certain external factor, a

variety of logic gates will be realized with dynamic controllability ... Meanwhile, if Φ can be controlled by some intrinsic dimensions of the photonic state, the logical rotation gate will be analogous to the CNOT gate, one fundamental component of quantum logic circuits. Such control mode would pave the way for a complete logic network of non-separable states' in **Line 99-106**.

- (3) Some descriptions about the controlling mode of the CLC device have been revised as 'whose functionality can be flexibly controlled by the relative angle between the CLC device and the incident light. One straightforward control mode is through the mechanical rotation of the CLC device ... whose logical rotation angle Φ can be tuned in a wide range, potentially acting as various logic gates' in **Line 133-137**.
- (4) Additional results of the device fabrication and the large aperture device have been added as 'and their homogeneous planar textures are shown in Fig. 2e and Fig. S5, whose functional region can be scaled up to about 5.8 cm^2 ' in **Line 165-166**, and added in the revised **Fig. S5**.
- (5) The descriptions about the transmittance spectra have been revised as 'Comparing the two points labelled by horizontal and vertical drop lines in Fig. 3c, when θ grows from 0° to 14° , the increment of ϕ_{out} at 632.8 nm approaches 90° with the $\text{PFM} > 0.90$, indicating that the polarization rotation can be flexibly tuned by altering the incident angle' in **Line 179-181**.
- (6) The new results of the transmission efficiency have been added as 'The transmission efficiency is about 80% and can be further improved to over 90% using anti-reflection substrates' in **Line 224-225**, and added as **Fig. S7**.
- (7) The new results of the logical rotation of higher dimensional non-separable states have been added as 'Not limited to $l = +1$, we also execute the logical rotation of the non-separable states with $l = -1$ and $l = +4$ (Fig. S1b,c), and obtain a similar evolution trajectory on corresponding HOPSs (Fig. S8), which indicates the universal applicability of the proposed CLC device' in **Line 225-227**, and added as **Fig. S8**.
- (8) The principles and applications of the proposed angular motion tracking have been explained more clearly as 'In the above implementation, the logical rotation is controlled by the tilt angle θ_{mech} of the device (an external mechanical quantity), and such control mode is referred to as *mechanical mode* (Fig. 6a). θ_{mech} determines the output non-separable state as shown in Fig. 5b-e, which is characterized by the orientation angle β of the dark fringe segmenting the analyzed intensity profile (i.e., the intensity minima). β can be extracted via the digital image

processing (Fig. S9) ... Based on the collected intensity profiles and the working curve' in **Line 252-259**, and added in the revised **Fig. 6**.

- (9) The results and discussions about the logical rotation controlled by the photonic wave vector, and the potential logic network, have been added as 'A step from the single CLC logical rotation gate towards the complete logic network is vital for developing algorithms or protocols based on non-separable states. Accordingly, the logical rotation needs to be controlled by a photonic state, and the output state of the former stage needs to be inserted into the control and the input of the successive logical rotation gate (see detailed discussions in Supplementary Note 5 and Note 6). To satisfy these two requirements, on one hand, we introduce the photonic wave vector direction θ_{phot} , an intrinsic DoF of light, as another control method of the logical rotation (Fig. 6d), and name it *wave vector mode* distinguished to the previous *mechanical mode* (see detailed analysis in Supplementary Note 5 and Fig. S10). Considering the relation $\theta = \theta_{\text{mech}} - \theta_{\text{phot}}$, the altering θ_{phot} acts as the control, while the preset constant θ_{mech} plays the role of 'bias' to offer the desired operating region of the logical rotation. A circuit (Fig. S11) to validate the *wave vector mode* is executed by the setup in Fig. 6e. Through the equal-energy distributed diffraction from a Dammann grating, the initial non-separable state with radial polarization is simultaneously encoded with $\theta_{\text{phot}} = -2.34^\circ, -0.78^\circ, +0.78^\circ, \text{ and } +2.34^\circ$ in four diffracted channels (Methods). For the CLC logic gate tilted to a fixed $\theta_{\text{mech}} = 17.84^\circ$, different intensity profiles of the analyzed outputs (Fig. 6f and Fig. S11) indicate that the logical rotation is efficiently controlled by the photonic wave vector. On the other hand, a reverse-controlling unit is demanded to reversely control θ_{phot} based on the non-separable state. CLC logical rotation gates and reverse-controlling units may assemble sequentially into a complete logic network (see configuration in Supplementary Note 6 and Fig. S12). In the proof-of-principle experiment, we consider the reduced non-separable states with $l = 0$ (i.e., the spin cebits), and a simplified logic network has been demonstrated with additional easily-accessible optical elements (Fig. S12). For more general non-separable states, versatile metasurfaces capable of manipulating multiple VBs are promising candidates for the reverse-controlling unit. The composed functional logic network would have great potentials in high-dimensional photonics and optical informatics' in **Line 263-283**, and added in the revised **Fig. 6**. Related results have also been added in **Supplementary Note 5, 6** and **Fig. S10-S12**.
- (10) The results and discussions about the main structural factors affecting the performance of the logical rotation have been added as 'Main structural factors affecting the performance of the logical rotation have been deeply investigated,

including the interaction length L , the ratio of operating wavelength λ to the short-wavelength edge of PBG n_{op} , and the chirality of CLCs. Firstly, sufficient L is necessary for the functionality of the logical rotation gate, as proved by the highly-restricted performance of a much thinner device in Fig. S13. Secondly, when $\lambda/(n_{op})$ increases, λ gets closer to the PBG and the dispersion of optical rotatory power becomes stronger. Therefore, the logical rotation angle Φ depends more sensitively on θ , while their dynamic ranges are also affected (Fig. S14). Thirdly, a CLC device with the opposite chirality leads to an opposite direction of the logical rotation (Fig. S15). Additionally, opposite incident angle $-\theta$ is equivalent to $+\theta$ because of the system symmetry. In a word, for a specific application, a competent CLC logic gate can be carefully optimized in the light of above principles, and the intended chiral superstructures can be fabricated readily with proper type and concentration of chiral dopants, thanks to their intriguing self-assembly' in **Line 296-306** in the section of **Discussion**, and added as **Fig. S13-15**.

- (11) The analysis about light-controlled logical rotation has been added as 'Moreover, if the fantastic stimuli-responsive CLC materials are adopted, the dynamic logical rotation controlled by light or heat could be rationally anticipated' in **Line 306-307** in the section of **Discussion**, and added as **Fig. S16**.
- (12) The discussions about the logical rotation gate have been revised as 'which contributes to angular motion tracking in the field of metrology and potential logic network of non-separable states ... our scheme enables a much wider tuning range and simpler configuration. Although angle dependence usually manifests in the undesirable form of energy loss or aberration for traditional devices, here we unlock the incident angle as an extra DoF for dynamic controllability. Such control principle averts repetitive re-organization of the internal nanostructures and exhibits merits of synchronous response and high reliability' in **Line 310-315** in the section of **Discussion**.
- (13) Some descriptions about the **Methods** have been added as 'While for the comparative experiment of larger $\lambda/(n_{op})$ in Fig. S14, the dopant ratio was 2.28 wt%. The right-handed CLC in Fig. S15 was fabricated by mixing E7 and right-handed chiral dopant R5011 (HCCH, China)' in **Line 329-330**, and as 'The other thin CLC cell for comparative experiment was formed by 9.5 μm spacers. ITO glass substrates of $3.0 \times 3.0 \text{ cm}^2$ were used to get larger aperture in Fig. S5. Glass substrates with anti-reflection coating were used in Fig. S7' in **Line 338-340**, and as ' $q = -0.5$ and $q = +2$ for Fig. S8 ... by a 1×4 Dammann grating, the initial radially polarized VB was diffracted into four equal-energy orders with the same non-separable state but different propagation directions (i.e., the wave vectors). In

the case of normal incidence, the grating equation is expressed by $\sin\theta_{\text{phot}} = K\lambda/\Lambda_{\text{DG}}$, where $K = \pm 1$ and ± 3 , θ_{phot} is the direction of respective output wave vector, $\lambda = 632.8$ nm, and $\Lambda_{\text{DG}} = 46.5$ μm is the grating period. All θ_{phot} were measured by triangle relation' in **Line 388-393**.

- (14) Some descriptions about the non-separable states have been added in **Supplementary Note 1**. Some explanations about the eigenmodes in CLCs have been added in **Supplementary Note 4**. We have also added **Supplementary Note 5** to discuss the logical rotation controlled by the photonic wave vector, and added **Supplementary Note 6** to discuss the logic network of non-separable states and its demo for spin cebits.
- (15) **Figure 2, 3, 5, 6** have been partially/wholly revised. **Figure S1, S5, S7, S8, S10-S12, S14-S16** have been added in the revised Supplementary Information. The related **figure captions** have also been revised/added.
- (16) The references have been renewed. In the revised manuscript, related references have been added as **References** [19, 42-45]. In the revised Supplementary Information, new references have been added as **Supplementary Information References** [21-32].
- (17) We have checked the statements required by the editorial policies. We have also carefully proofread the whole revised manuscript.

REVIEWER COMMENTS

Reviewer #1 (Remarks to the Author):

In the revised version of the article "Logical rotation of non-separable states via uniformly self-assembled chiral superstructures" by Yi-Heng Zhang et al., the authors have extensively addressed my previous concerns about whether the proposed device was actually performing "logical operations".

In particular, the authors have cleverly noticed that the propagation angle of light, which acts equivalently to the system's mechanical rotation, is both a control and an output logical variable, provided that appropriate extra elements are included in their photonic network. Clearly, this increases the complexity of the system, which results bulky even for a modest number of network nodes. Nonetheless, this drawback might be mitigated by the potential applications that could stem from the non-standard logic rules intrinsic to the classical non-separable states that are here being manipulated.

At the present stage, I would hence endorse publication of the manuscript in Nature Communications, provided that the authors clearly identify what are the algorithms and the computational concepts that would benefit from a logic network based on classical non-separable states, keeping into account that - as appropriately pointed out by the author themselves - certain key features of true quantum system such as nonlocality are here absent.

Reviewer #2 (Remarks to the Author):

The paper has been well revised, making clear points, especially taking insight into logical rotation of non-separable states with self-assembled chiral superstructures. It's ready for publication.

Reviewer #3 (Remarks to the Author):

I am happy to see this revision effort and believe the manuscript has been largely improved, especially more theoretical analysis and solid experiments are added. I would therefore recommend its publication.

RESPONSE TO THE REVIEWERS:

We greatly appreciate all the reviewers for accessing our revised manuscript of NCOMMS-23-22930A, and providing valuable comments. All reviewers' comments and suggestions are quite helpful for us to substantially improve the quality and readability of this manuscript.

In the following, we provide a point-to-point reply to all reviewers' comments. A list of changes is attached in the end, and all changes are highlighted in blue in the revised manuscript and supplementary information. We hope the new version would be found more convincing and meet the journal's criteria for publication.

We truly appreciate all Reviewers' valuable comments and kind recommendations.

Reviewer #1

In the revised version of the article "Logical rotation of non-separable states via uniformly self-assembled chiral superstructures" by Yi-Heng Zhang et al., the authors have extensively addressed my previous concerns about whether the proposed device was actually performing "logical operations".

In particular, the authors have cleverly noticed that the propagation angle of light, which acts equivalently to the system's mechanical rotation, is both a control and an output logical variable, provided that appropriate extra elements are included in their photonic network. Clearly, this increases the complexity of the system, which results bulky even for a modest number of network nodes. Nonetheless, this drawback might be mitigated by the potential applications that could stem from the non-standard logic rules intrinsic to the classical non-separable states that are here being manipulated.

At the present stage, I would hence endorse publication of the manuscript in Nature Communications, provided that the authors clearly identify what are the algorithms and the computational concepts that would benefit from a logic network based on classical non-separable states, keeping into account that - as appropriately pointed out by the author themselves - certain key features of true quantum system such as nonlocality are here absent.

Response

We truly appreciate the respected Reviewer #1 for his or her precious time in reading our revised manuscript, recognizing our revision effort and providing positive comments. We are very grateful that in the previous report, Reviewer #1 raised the insightful concerns on whether the proposed device was actually performing the logical operation, and provided very professional summary of the fundamental requirements of a logic control system. Reviewer #1's thoughtful comments and suggestions have inspired us to further consider the equivalence between the propagation angle of light and the mechanical rotation of the system, so that a logic network could be established with appropriate extra elements. We really appreciate all previous valuable comments, which are greatly helpful to improve the quality and significance of our work.

At the present stage, Reviewer #1 raises the insightful remarks that the system complexity is increased, and this drawback could be mitigated by some potential applications from the non-standard logical rotation rules of the classical non-separable states. We totally agree with Reviewer #1's concerns on this relatively bulky system. In this revision, we have carefully considered the potential applications, and devote effort to identifying the potential computation concepts which may benefit from a logic network based on classical non-separable states. For your kind reference, we have prepared a schematic illustration of the related concepts in Fig. R1.

Fig. R1 | Schematic illustration of the analogous version of quantum walks via proposed logic network.

One potential application of this proposed logic network is the analogous version of quantum walks (Fig. R1), which are instrumental computation tools for studying phenomena in condensed matters [*Optica*, **7**, 108-114 (2020); *Optica*, **10**, 324-331 (2023)]. As an exemplary protocol, the lattice in common quantum walks would be encoded in the wave vector DoF, and the coin would be encoded into the non-separable state. Correspondingly, the coin rotation can be implemented by the CLC logical

rotation gates, while the walker translation can be realized by the reverse-controlling units. These contribute to an analogous version of quantum walk based on classical non-separable states. When the non-separable state is reduced to the spin cebit (i.e., polarization), this elementary case can also be realized by a stack of wave plates and polarization gratings, and used to simulate a Chern insulator [*Optica*, 7, 108-114 (2020)]. Compared to spin cebits, non-separable states are associated with both spin and orbital DoFs, so we can encode the coin into more complex form. In addition, the non-trivial rules of the logical rotation allow us to design the coin rotation in a highly flexible way. These might open new avenues for modeling complicated physical systems.

Moreover, in a more general view, the proposed logic network here is a kind of optical computing system where parallel computation can be executed in the speed of light propagation. A rich set of response functions can be rationally constructed from the non-standard logical rotation rules, which are determined by some specific factors including the pitch and the chirality of the CLC material and the preset tilt angle of the CLC device. Notably, a non-separable cebit offers a much larger state space than a traditional bit, and this would greatly extend the information capacity. We anticipate that such a functional logic network, established on the CLC logical rotation gates and other versatile optical elements, would unveil new opportunities of non-separable states in high-dimensional photonics and optical informatics.

IN ALL, we sincerely thank the respected Reviewer #1 for recognizing our effort and improvement in addressing previous concerns. Following Reviewer #1's insightful suggestion in this report, we have carefully considered the potential applications in analogous quantum walks and optical computing based on this logic network, which are greatly helpful to further improve the quality and significance of this work. Corresponding discussions have been added in **Line 282-286** in the revised manuscript and **Supplementary Note 6**. We really hope the new version would be found more convincing, and suitable for publication now.

Reviewer #2

The paper has been well revised, making clear points, especially taking insight into logical rotation of non-separable states with self-assembled chiral superstructures. It's ready for publication.

Response

We greatly appreciate the respected Reviewer #2 for his or her precious time in assessing our revised manuscript, providing very positive comments, and kind recommendation for its publication.

Reviewer #3

I am happy to see this revision effort and believe the manuscript has been largely improved, especially more theoretical analysis and solid experiments are added. I would therefore recommend its publication.

Response

We truly appreciate the respected Reviewer #3 for his or her precious time in assessing our revised manuscript, providing very positive comments, and kind recommendation for its publication.

LIST OF CHANGES

- (1) Some discussions about the computation concepts which may benefit from the logic network based on non-separable states have been added as ‘The composed functional logic network would have great potentials in high-dimensional photonics, optical computing and high-capacity optical informatics. In particular, it might benefit some computation concepts in the analogous version of quantum walks, since the non-separable-states-mediated coins could allow more complex form and more flexible coin rotation rules in this logic network (see more discussions in Supplementary Note 6).’ in **Line 282-286**, and added as ‘One potential application of this proposed logic network is the analogous version of quantum walks, which are instrumental computation tools for studying phenomena in condensed matters. As an exemplary protocol, the lattice in common quantum

walks would be encoded in the wave vector DoF, and the coin would be encoded into the non-separable state. Correspondingly, the coin rotation can be implemented by the CLC logical rotation gates, while the walker translation can be realized by the reverse-controlling units. These contribute to an analogous version of quantum walk based on classical non-separable states. When the non-separable state is reduced to the spin cebit (i.e., polarization), this elementary case can also be realized by a stack of wave plates and polarization gratings, and used to simulate a Chern insulator. Compared to spin cebits, non-separable states are associated with both spin and orbital DoFs, so we can encode the coin into more complex form. In addition, the non-trivial rules of the logical rotation allow us to design the coin rotation in a highly flexible way. These might open new avenues for modeling complicated physical systems. Moreover, in a more general view, the proposed logic network here is a kind of optical computing system where parallel computation can be executed in the speed of light propagation. A rich set of response functions can be rationally constructed from the non-standard logical rotation rules, which are determined by some specific factors including the pitch and the chirality of the CLC material and the preset tilt angle of the CLC device. Notably, a non-separable cebit offers a much larger state space than a traditional bit, and this would greatly extend the information capacity. In all, we anticipate that such a functional logic network, established on the CLC logical rotation gates and other versatile optical elements, would unveil new opportunities of non-separable states in high-dimensional photonics, optical computing and optical informatics.’ in **Supplementary Note 6**.

- (2) The references have been renewed. In the revised manuscript, related references have been added as **References** [46,47]. In the revised Supplementary Information, new references have been added as **Supplementary Information References** [31-33].
- (3) We have checked the statements required by the editorial policies. We have also carefully proofread the whole revised manuscript.

REVIEWERS' COMMENTS

Reviewer #1 (Remarks to the Author):

after considering the clarifications provided in the last version of the manuscript, I believe that its scientific level is now suitable for publication in Nature Communications.

RESPONSE TO THE REVIEWERS:

We greatly appreciate all the reviewers for accessing our revised manuscript of NCOMMS-23-22930B, and providing valuable comments. We are truly grateful to all Reviewers' valuable comments and kind recommendations.

Reviewer #1

Dear Authors, dear Editor,

after considering the clarifications provided in the last version of the manuscript, I believe that its scientific level is now suitable for publication in Nature Communications.

Response

We truly appreciate the respected Reviewer #1 for his or her precious time in assessing our revised manuscript, recognizing our revision effort, and kind recommendation for its publication. We sincerely thank Reviewer #1's insightful comments and suggestions throughout the entire revision process, which are greatly helpful to significantly improve our work.